# Structural basis of fast N-type inactivation in $K_v$ channels

Xiao-Feng Tan[1,4✉], Ana I. Fernández-Mariño[1,2,4✉], Yan Li[3], Tsg-Hui Chang[1] & Kenton J. Swartz[1✉]

Action potentials are generated by opening of voltage-activated sodium ($Na_v$) and potassium ($K_v$) channels[1], which can rapidly inactivate to shape the nerve impulse and contribute to synaptic facilitation and short-term memory[1–4]. The mechanism of fast inactivation was proposed to involve an intracellular domain that blocks the internal pore in both $Na_v$[5,6] and $K_v$[7–9] channels; however, recent studies in $Na_v$[10,11] and $K_v$[12,13] channels support a mechanism in which the internal pore closes during inactivation. Here we investigate the mechanism of fast inactivation in the Shaker $K_v$ channel using cryo-electron microscopy, mass spectrometry and electrophysiology. We resolved structures of a fully inactivated state in which the non-polar end of the N terminus plugs the internal pore in an extended conformation. The N-terminal methionine is deleted, leaving an alanine that is acetylated and interacts with a pore-lining isoleucine residue where RNA editing regulates fast inactivation[14]. Opening of the internal activation gate is required for fast inactivation because it enables the plug domain to block the pore and repositions gate residues to interact with and stabilize that domain. We also show that external $K^+$ destabilizes the inactivated state by altering the conformation of the ion selectivity filter rather than by electrostatic repulsion. These findings establish the mechanism of fast inactivation in $K_v$ channels, revealing how it is regulated by RNA editing and N-terminal acetylation, and providing a framework for understanding related mechanisms in other voltage-activated channels.

Hodgkin and Huxley described the voltage-activated conductances for $Na^+$ and $K^+$ that enable neurons to generate action potentials[1]. These conductances arise from voltage-activated cation channels that are selective for either $Na^+$ ($Na_v$) or $K^+$ ($K_v$). Both $Na_v$ and $K_v$ channels contain voltage-sensing domains that activate when the internal side of the cell membrane becomes less negative (depolarized), leading to opening of the activation gate at the internal end of the pore[10,15–17]. $Na_v$ channels and some types of $K_v$ channels also rapidly inactivate on the millisecond timescale in response to sustained depolarization to shape the action potential and serve as a form of short-term memory[1–4]. Classical studies on fast inactivation in $Na_v$ channels envisioned a tethered intracellular domain blocking the pore[5,6]. Subsequent studies on $Na_v$ channels identified a domain containing Ile, Phe and Met residues (termed the IFM motif) in the linker between the third and fourth repeats as a potential tethered blocker[18,19]. However, the IFM motif has recently been shown to bind outside the pore[10], and inactivation has been proposed to result from closure of the internal pore rather than a pore-blocking mechanism[11,12].

In the Shaker $K_v$ channel from *Drosophila melanogaster*, N-type inactivation occurs on a similar timescale to inactivation in $Na_v$ channels, and studies have supported a mechanism in which the N terminus of the protein blocks the pore[7,8,20–25]. For mammalian $K_v$1 channels related to Shaker, the proposed tethered blocker can also be supplied by cytoplasmic regulatory β-subunits[26], where fast inactivation is critical for synaptic plasticity[3,4]. The importance of N-type inactivation for electrical signalling is reinforced by its exquisite regulation by RNA editing[14] and membrane lipids such as phosphatidylinositol 4,5-bisphosphate[27]. In humans, mutations in $K_v$1 channels that disrupt fast inactivation cause episodic ataxia[28,29]. A related mechanism of fast N-type inactivation occurs in $K_v$3.4 channels where phosphorylation of the N terminus regulates inactivation[9]. In large conductance calcium-activated potassium ($BK_{Ca}$) channels, the proposed tethered inactivation particle is located within auxiliary β-subunits[30,31] and inactivation is regulated within the superchiasmatic nucleus to set the diurnal variations in excitability underlying the circadian rhythm[32]. Despite the importance of rapid inactivation in $K_v$ and $BK_{Ca}$ channels, its structural basis remains to be elucidated.

In the present study, we investigated the mechanism of fast inactivation in the Shaker $K_v$ channel by solving structures under conditions that favour the inactivated state, characterizing the composition of the N terminus using mass spectrometry and validating the emerging mechanism using electrophysiology. Our findings reveal how the immediate non-polar end of the N terminus plugs the pore in an extended conformation to block ion conduction, and we identified several key regions where interactions between the N terminus and the pore stabilize the inactivated state. We also found that the N terminus of Shaker is modified by removal of the initiator Met and acetylation of the remaining N-terminal Ala, which interacts directly with a pore-lining residue where RNA editing regulates fast inactivation[14]. The absence of positive

[1]Molecular Physiology and Biophysics Section, Porter Neuroscience Research Center, National Institute of Neurological Disorders and Stroke, National Institutes of Health, Bethesda, MD, USA. [2]Department of Physiology and Biophysics, University of Colorado, Anschutz Medical Campus, Aurora, CO, USA. [3]NINDS Proteomics Core Facility, Porter Neuroscience Research Center, National Institute of Neurological Disorders and Stroke, National Institutes of Health, Bethesda, MD, USA. [4]These authors contributed equally: Xiao-Feng Tan, Ana I. Fernández-Mariño. ✉e-mail: xiaofeng.tan@nih.gov; ana.fernandez-marino@cuanschutz.edu; swartzk@ninds.nih.gov

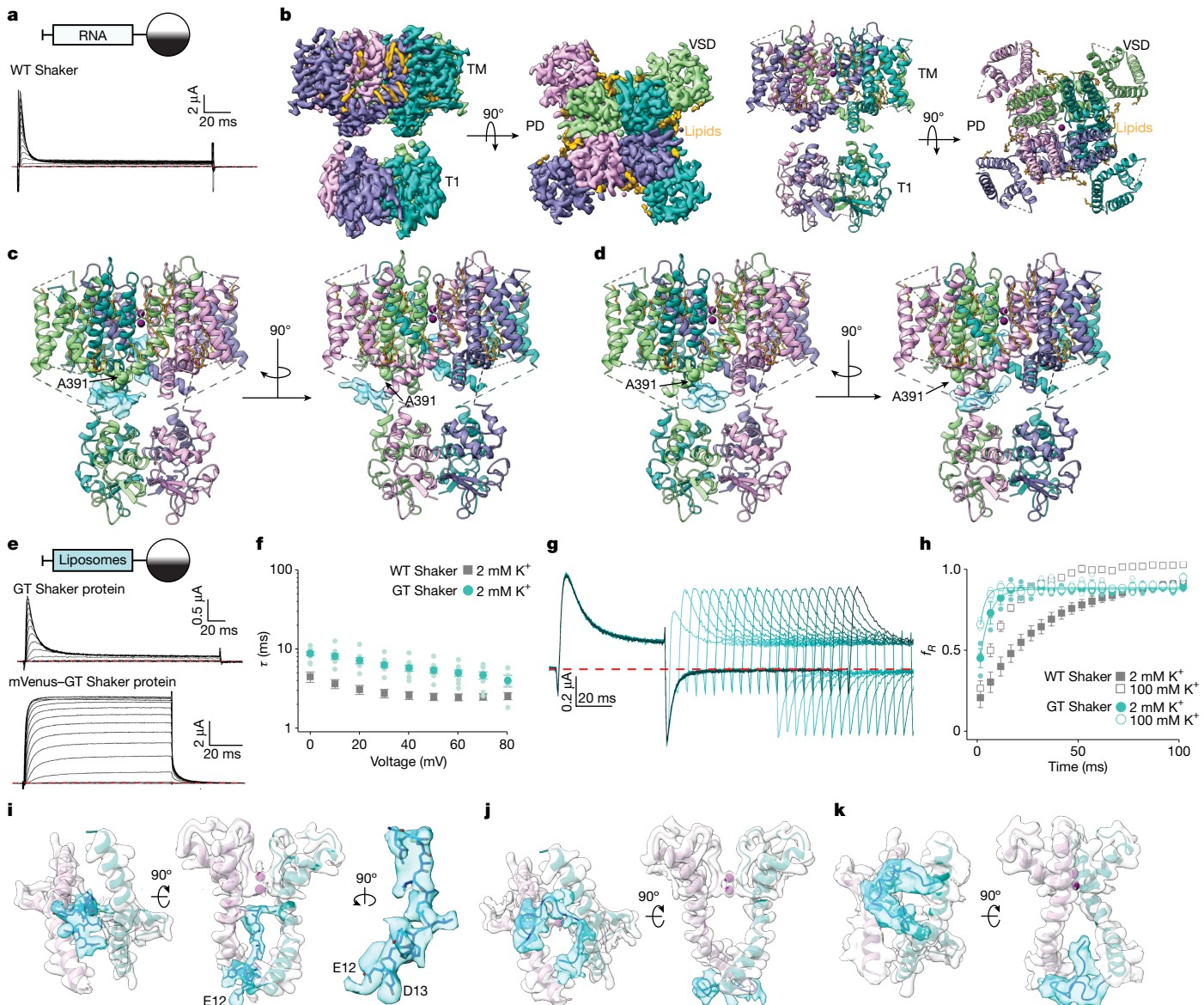

**Fig. 1 | Structures of modified Shaker $K_v$ channels. a**, Current traces recorded from an RNA-injected oocyte using 2 mM external $K^+$. The holding voltage was −100 mV, with 10-mV steps up to +80 mV. The red dashed line is zero current. **b**, Side and external views of the cryo-EM C4 map and model for proteolysed samples of C-terminally tagged WT Shaker. PD, poredomain; TM, transmembrane; VSD, voltage-sensing domain. **c,d**, Side views of the class A (**c**) and class B (**d**) models for proteolysed samples of C-terminally tagged WT Shaker with C1 symmetry. The light blue density in the chamber between the transmembrane and T1 domains and within the pore is not seen with C4 symmetry. **e**, Current traces for Shaker expressed in oocytes by injecting proteoliposomes. The same protocol as panel **a** with a tail voltage of −50 mV. P/−4 subtraction was used. **f**, Time constants of fast inactivation ($\tau$) versus test voltage for WT Shaker using RNA injection compared with GT Shaker using proteoliposome injection. $\tau$ were obtained from single exponential fits to current traces ($n = 7$ cells in 4 independent experiments for WT Shaker and $n = 8$ cells in 5 independent experiments for GT Shaker). **g**, Current traces measuring recovery from inactivation for GT Shaker using proteoliposome injection. The holding voltage was −100 mV, with initial 60-ms steps to +30 mV followed by varying time for recovery at −100 mV (initially 2 ms and increased in 5-ms intervals) before eliciting a second step to +30 mV. External $K^+$ was 100 mM. P/−4 subtraction was used, and the red dashed line denotes zero current. **h**, Fraction of current recovered (fR) versus time for GT Shaker using proteoliposome injection ($n = 3$ cells in 2 independent experiments for 2 mM external $K^+$ and $n = 3$ cells in 2 independent experiments for 100 mM external $K^+$) compared with WT Shaker using RNA injection ($n = 7$ cells in 4 independent experiments for 2 mM external $K^+$ and $n = 5$ cells in 4 independent experiments for 100 mM external $K^+$) using the protocol in panel **g**. See Extended Data Fig. 4a,b for traces for WT Shaker. **i–k**, Side and internal views of the cryo-EM C1 maps and models for classes A (**i**), B (**j**) and C (**k**) of GT Shaker. Extra N-terminal density in the internal pore is shown in light blue. Error bars are s.e.m. (**f,h**).

charges within the N-terminal plug domain led us to re-examine how external $K^+$ destabilizes the inactivated state, revealing that the ion selectivity filter within the external pore regulates fast N-type inactivation.

## Structures of the N-type-inactivated state of Shaker

We expressed the full-length wild-type (WT) Shaker $K_v$ channel that rapidly inactivates[7,8] (Fig. 1a) along with a C-terminal mVenus tag in mammalian cells, purified and reconstituted the protein into lipid nanodiscs using MSP1E3D1 (ref. 33) in 150 mM K+, and then exchanged the solution to one containing 4 mM $K^+$ and 46 mM $Na^+$, ionic conditions that favour inactivation[21]. After removal of the mVenus tag, we used cryo-electron microscopy (cryo-EM) to evaluate the structure under these conditions. After refinement using C4 symmetry, the cryo-EM data enabled us to determine the structure of the WT Shaker $K_v$ channel at 2.73 Å resolution, which could be further improved by

separately refining the transmembrane domain to 2.67 Å and the cytoplasmic T1 domain to 2.98 Å (Extended Data Table 1, Extended Data Fig. 1 and Supplementary Fig. 1). These maps were of similar quality throughout, sufficient for de novo model building, with clearly resolved densities for many side chains (Fig. 1b). The peripheral S1–S4 voltage-sensing domains were in an activated conformation, the internal pore was open and the ion selectivity filter in the external pore adopts a dilated C-type-inactivated conformation, similar to earlier structures of Shaker[34,35]. However, no density within the internal pore could readily be attributed to the N terminus within cryo-EM maps for the WT channel when imposing C4 symmetry (Fig. 1b, Extended Data Fig. 1 and Supplementary Fig. 1). After focused classification imposing C1 symmetry, we observed three classes (A–C; Fig. 1c,d and Supplementary Fig. 1), two of which (A and B) contained extra density inside the internal pore or outside the pore near the window created by the linkers between the cytoplasmic T1 domain and the S1 transmembrane helices (Fig. 1c,d). The extra densities in class A and B were not continuous, and no additional density was evident within the internal pore in class C (representing 50% of particles across the three classes; Supplementary Fig. 1), leading us to suspect that the N termini of Shaker had been proteolysed during sample preparation. Although these additional densities probably originate from the N terminus, they were not of sufficient resolution to model this region of the protein.

To mitigate proteolysis of the N terminus, we expressed a construct containing mVenus and the affinity purification tag on the N terminus, enabling us to purify the protein with at least one N terminus intact within individual tetramers. Subsequent removal of the N-terminal mVenus tag using TEV protease leaves two additional (Gly and Thr) residues on the N terminus, which we confirmed with mass spectrometry (Extended Data Fig. 2a,b). To test whether the additional GT residues interfere with rapid inactivation, we injected *Xenopus* oocytes with purified GT Shaker protein that had been reconstituted into liposomes (see Methods) and used voltage-clamp techniques to characterize its inactivation properties. The kinetics for the onset of inactivation in the GT Shaker $K_v$ channel were slower than the WT full-length channel expressed following RNA injection (Fig. 1a,e,f), whereas recovery from inactivation at negative membrane voltages was faster than the WT channel (Fig. 1g,h), suggesting that the two additional residues do not interfere with inactivation but probably destabilize the inactivated state. Injection of the purified construct reconstituted into liposomes without previous cleavage of the N-terminal mVenus tag eliminated fast inactivation (Fig. 1e), indicating that the bulky fluorescent protein on the N terminus is sufficient to disrupt N-type inactivation.

To capture the inactivated state of GT Shaker, we reconstituted the protein into nanodiscs and were able to obtain randomly oriented particles for cryo-EM imaging (Extended Data Figs. 1 and 3). However, when applying C1 symmetry without an appropriate mask, the fourfold symmetric transmembrane domains disrupted the classification of different conformations of the N-terminal peptide, particularly the region of the peptide within the internal pore located along the central axis of the C4 symmetric transmembrane domain. This led to the averaging of the N-terminal peptide density within the internal pore into a featureless, bulky ball-like density. In addition, the rest of the N-terminal peptide in the chamber between the transmembrane and T1 domains was eliminated (Extended Data Fig. 3). We therefore used a cylindrical mask covering an asymmetric region of the protein to initially obtain a low-resolution map containing an L-shaped density in the region of interest and then used an L-shape mask during further 3D classification to obtain three classes (A–C) that contained additional density within the pore or chamber (Extended Data Tables 1 and 2 and Extended Data Fig. 3). In each of these cryo-EM maps, the voltage-sensing domains are activated and the internal S6 gate open. In class A resolved to 2.94 Å, the additional density fully plugs the internal pore in an extended conformation (Fig. 1i); in class B, the additional density is positioned at the internal end of the pore (Fig. 1j); and in class C, the additional density begins to enter the internal pore (Fig. 1k). Although classes B and C may represent intermediate states during inactivation, we also cannot exclude the possibility that they arise from tetramers containing truncated N termini resulting from proteolytic cleavage. We tentatively conclude that class A corresponds to a fully inactivated state in GT Shaker because the non-polar region of the N terminus inserts deeply into the internal pore with the N-terminal Met (now at position 3) interacting directly with I470 (Fig. 1i; also see Fig. 3a and text below). Density corresponding to the additional two residues present in this construct is visible within class A, curling towards the intracellular end of the pore to form an inverted J shape (Fig. 1i and Extended Data Fig. 3). The plug domain adopts an extended conformation, consistent with earlier functional studies proposing that a single N terminus enters the pore in an extended conformation to inactivate $K_v$ channels[20,23,36].

In attempting to solve structures of the N-type-inactivated state without the additional GT residues, we returned to expression of the C-terminally tagged channel while using more stringent conditions to inhibit proteases during purification and succeeded in obtaining samples that by mass spectrometry had not been proteolysed (Extended Data Fig. 2c,d). However, in these samples, we discovered that the N-terminal Met had been consistently cleaved and was modified by acetylation at the A2 position (Fig. 2a–c and Extended Data Fig. 2c,d), a common modification of membrane proteins produced by N-terminal acetyltransferases[37] proposed to shield proteins from degradation[38]. To determine whether this truncated and acetylated channel corresponds to the channel that has been studied with functional approaches, we used the same purified AcA Shaker, reconstituted it into liposomes and injected those into oocytes. The kinetics for the onset and recovery from inactivation for the purified AcA Shaker construct were indistinguishable from the channel expressed using conventional approaches by injecting RNA for the full-length channels (Figs. 1a and 2d–g), suggesting that this modification is likely to be ubiquitous, occurring in both mammalian cells and *Xenopus* oocytes.

To increase our chances of obtaining high-resolution structures of the N-type-inactivated state, we introduced two mutations (E12K and D13K) that enhance inactivation (EI Shaker) by dramatically increasing the association rate for binding of the N-terminal inactivation peptide to the channel[21]. Analysis of this construct using mass spectrometry revealed cleavage of the N-terminal Met and acetylation of Ala2 (Extended Data Fig. 2e–g). We reconstituted purified AcA-EI Shaker into liposomes, injected those into oocytes and observed that the kinetics for the onset of inactivation were indeed faster than for the WT channel (Extended Data Fig. 4c,d). We collected a dataset for AcA-EI Shaker, and using similar data processing workflows to that for GT Shaker, we were able to resolve three classes of particles where the density of the N terminus was observed within the internal pore or chamber (Extended Data Tables 1 and 2, Extended Data Figs. 5 and 6 and Supplementary Fig. 2). In class A, we observed extra density in the chamber between the transmembrane and T1 domains (Extended Data Fig. 6a), similar to class B in GT Shaker (Fig. 1j). In class B, we observed two additional densities, one inserting deep into the pore and the other near to the lateral entrance to the chamber (Extended Data Fig. 6b,c). In class C resolved to 3.05 Å, we observed continuous density plugging the internal pore similarly to what we observed for class A in GT Shaker (Fig. 3d and Extended Data Fig. 6d,e). Although the density corresponding to the N terminus inside the pore was weaker than that for the surrounding pore domain, leaving open the possibility that the density arises from an average of alternative conformations, the N terminus adopts an extended conformation. Lowering the density threshold of the map for class C suggests that the N-terminal density was an average of two conformations: one dominant and one weaker (especially the region of the terminus containing relatively featureless sidechains: AcA-AAVAG). To address this challenge, we modelled the N-terminal density by fitting the first residue and those with big sidechains (L7 and Y8) before fitting

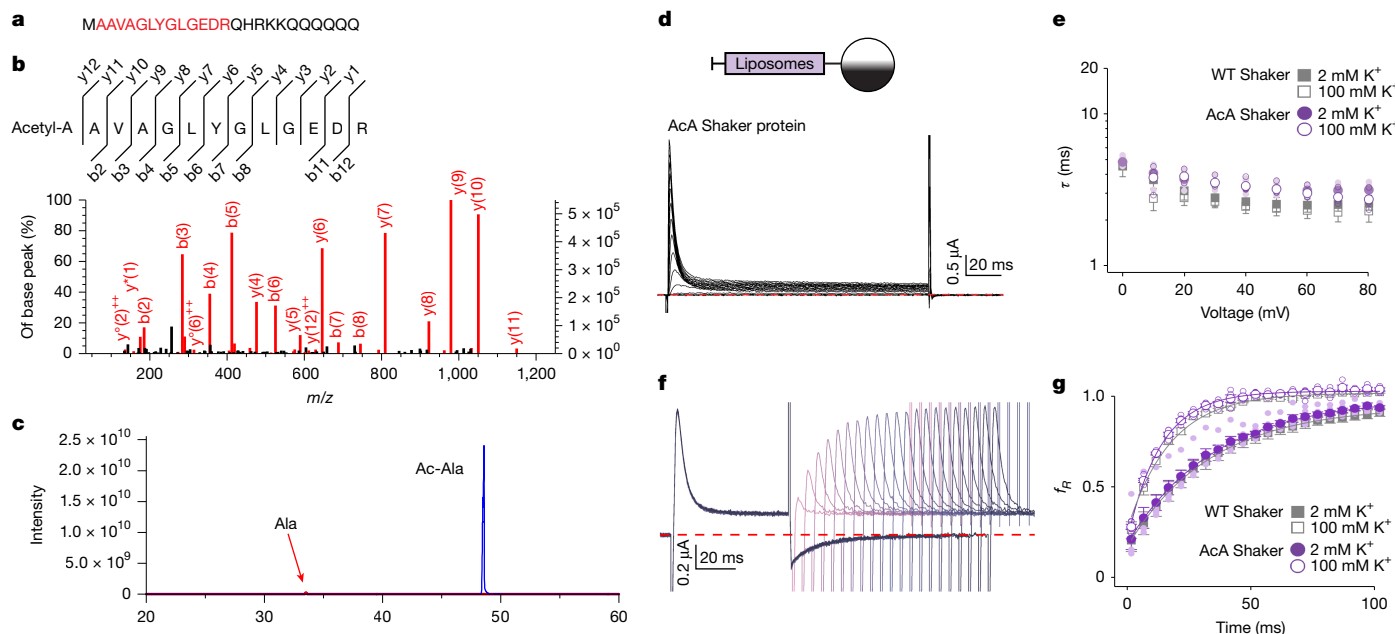

**Fig. 2 | Modifications of the N terminus in the full-length Shaker K$_v$ channel. a**, Sequence of the N terminus of Shaker, with the dominant species detected shown in red. **b**, MS/MS spectrum of the (Ac)AAVAGLYGLGEDR peptide. The left $y$ axis shows relative intensity and the right $y$ axis shows absolute intensity. Fragments matched to the theoretical masses are marked in red. **c**, Extracted ion chromatogram of (Ac)AAVAGLYGLGEDR (Ac-Ala; blue) and AAVAGLYGLGEDR (Ala; red). **d**, Current traces for AcA Shaker expressed in oocytes using proteoliposome injection. External K$^+$ was 2 mM. The holding voltage was −100 mV, with 10-mV steps up to +80 mV. P/−4 subtraction was used, and the red dashed line is zero current. **e**, Time constants of fast inactivation ($\tau$) versus test voltage for WT Shaker using RNA injection ($n = 7$ cells in 4 independent experiments for 2 mM external K$^+$ and $n = 5$ cells in 4 independent experiments for 100 mM external K$^+$) compared with AcA Shaker using proteoliposome

injection ($n = 7$ cells in 3 independent experiments for 2 mM external K$^+$ and $n = 5$ cells in 2 independent experiments for 100 mM external K$^+$). $\tau$ was obtained from single exponential fits to current traces. **f**, Family of current traces to measure recovery from inactivation for AcA Shaker expressed using proteoliposome injection. The holding voltage was −100 mV, with initial 60-ms steps to +30 mV, followed by varying time for recovery at −100 mV (initially 2 ms and increased in 5-ms intervals) before eliciting a second step to +30 mV. External K$^+$ was 100 mM. P/−4 subtraction was used, and the red dashed line is zero current. **g**, Fraction of current recovered (fR) versus recovery time for AcA Shaker using proteoliposome injection ($n = 6$ cells in 2 independent experiments for 2 mM external K$^+$ and $n = 5$ cells in 2 independent experiments for 100 mM external K$^+$) compared with WT Shaker using RNA injection (data from Fig. 1h) using the protocol shown in panel **f**. Error bars are s.e.m.

residues with short sidechains. We also observed extra density entering the chamber through the window formed by the linkers between the cytoplasmic T1 domain and the S1 transmembrane helices that we attribute to a second N terminus (Fig. 3d–f and Extended Data Fig. 6d,e). To validate our assignment of residues in the plug domain, we collected another dataset in the presence of an additional N-terminal peptide containing an acetylated Ala, increasing the number of particles in the fully inactivated state. In this case, we observed one main class (also 3.05 Å) in which the density for the N terminus within the pore was improved, making residue assignments more accurate, and we also observed an additional nearby density in the chamber corresponding to a second N-terminal peptide (Fig. 3g–i, Extended Data Figs. 5 and 7 and Supplementary Fig. 3).

The structures of GT Shaker and AcA-EI Shaker without or with free N-terminal peptide are similar in that at least one class in each contained continuous density fully plugging the internal pore with the end of the N terminus engaged with or nearby to I470 (Fig. 3a,d,g), a particularly critical position within the internal pore where RNA editing regulates fast inactivation[14]. These structures also show that the N terminus is stabilized within the pore by interactions between L7 and V478 and P475 in the S6 helices, forming a triad of close hydrophobic interactions (Fig. 3b,e,h). L7 has a particularly important role in stabilizing the inactivated state[20], supporting the importance of this hydrophobic triad. This triad is notable because P475 and V478 contribute to forming the activation gate when the channel is closed at negative voltages[16,39,40], and thus only becomes available for interacting with L7 after the internal gate opens. The cryo-EM map for AcA-EI Shaker with additional N-terminal peptide also has continuous density between V4 and V474

and P475 in S6, and between A5 and those same two residues in the S6 of another subunit (Fig. 3g), suggesting that those interactions stabilize the inactivated state. The cryo-EM maps for GT Shaker and AcA-EI Shaker without free peptide are also consistent with these interactions (Fig. 3a,d). These structures also show that Y8 in the N terminus, where mutations destabilize the inactivated state[20], is located near to N482 in S6 (Fig. 3c,i). The only charged residues resolved in our structures are those at positions 12 and 13, which in the WT protein are Glu and Asp and in our AcA-EI Shaker construct have been mutated to Lys. The density for these two residues in both GT Shaker and AcA-EI Shaker with additional free peptide are relatively strong, positioning them outside the pore within the chamber (Figs. 1i and 3d,g), consistent with mutations at these positions altering association rate between the N terminus and the pore through long-range electrostatic interactions without altering the stability of the inactivated state[20].

## Mechanism by which external K$^+$ promotes recovery

One of the key observations supporting a pore-blocking mechanism for N-type inactivation in Shaker is that raising the concentration of external K$^+$ and membrane hyperpolarization enhance the rate of recovery from N-type inactivation[24]. A seemingly related dependence on *trans* K$^+$ occurs for cationic pore-blocking scorpion toxins, in which binding of the toxin to the outer pore of K$^+$ channels is destabilized by raising the concentration of internal K$^+$ or membrane depolarization, manipulations that enhance K$^+$ occupancy of the pore to destabilize the cationic toxin through electrostatic repulsion[41–44]. Our structures, however, reveal that only non-polar residues enter the pore during inactivation

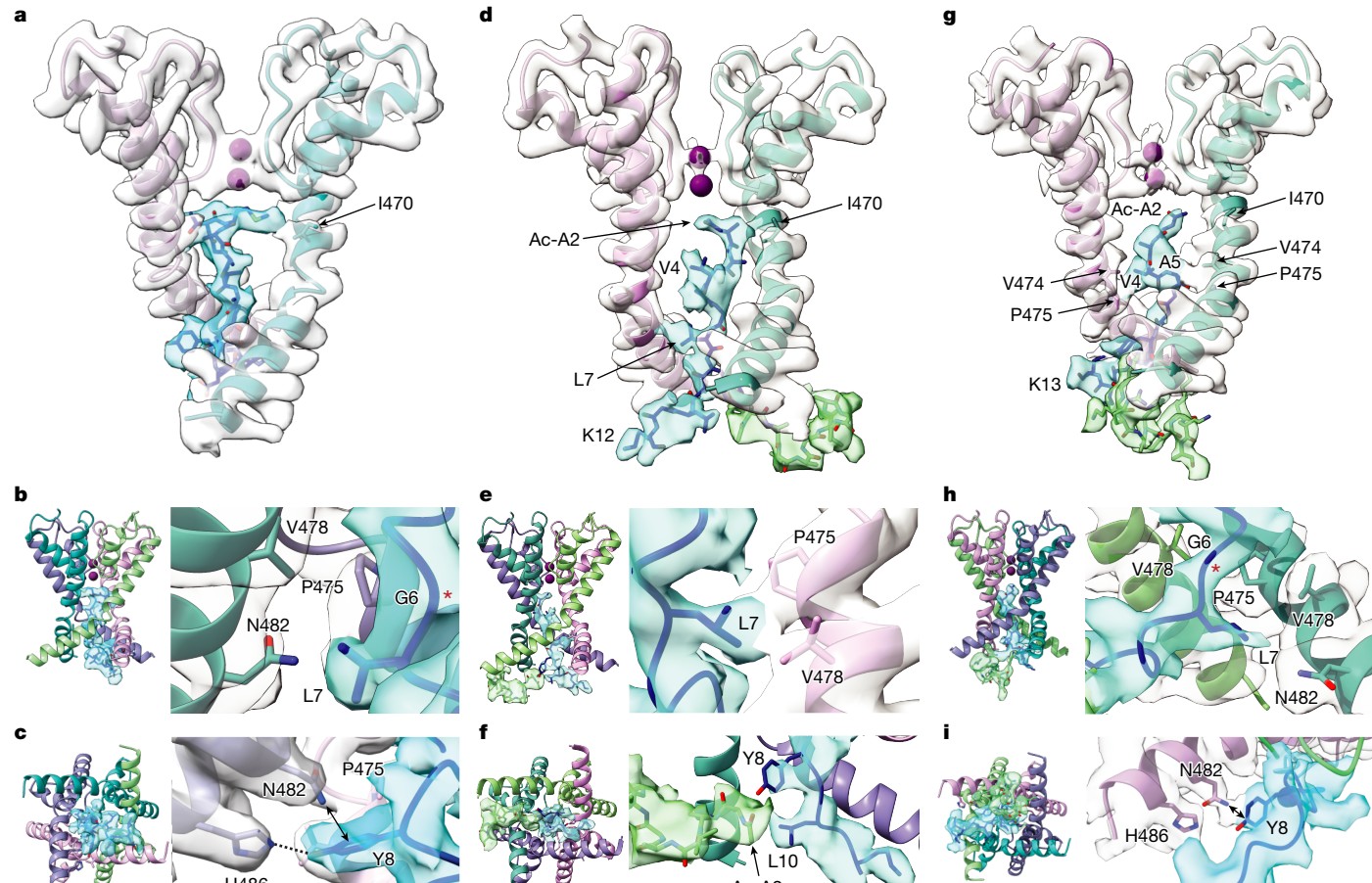

**Fig. 3 | Structures of the N-type-inactivated state of the Shaker K_v channel.**
**a**, Side view of the cryo-EM C1 map and model of the pore domain S6 helices from two opposing subunits for GT Shaker class A. **b**, Side views of the cryo-EM C1 map and model for GT Shaker class A, highlighting interactions between L7 in the N terminus and P475 and V478 in the S6 helix. Red asterisk highlights Cα of G6. **c**, Bottom views of the cryo-EM C1 map and model for GT Shaker class A, highlighting interactions between Y8 in the N terminus and N482 (arrow) and H486 (dashed line) in the S6 helix. **d**, Side view of the cryo-EM C1 map and model for AcA-EI Shaker class C. **e**, Side views of the cryo-EM C1 map and model for AcA-EI Shaker class C, highlighting interactions between L7 in the N terminus and P475 and V478 in the S6 helix. **f**, Bottom views of the cryo-EM C1 map and model for AcA-EI Shaker class C, highlighting interactions between Y8 in the N terminus and Ac-A2 from a second N-terminal peptide. **g**, Side view of the cryo-EM C1 map and model for the AcA-EI Shaker with added free N-terminal peptide. **h**, Side views of the cryo-EM C1 map and model for the AcA-EI Shaker with added free N-terminal peptide, highlighting interactions between L7 in the N terminus and P475 and V478 in the S6 helix. Red asterisk highlights Cα of G6. **i**, Bottom views of the cryo-EM C1 map and model for the AcA-EI Shaker with added free N-terminal peptide, highlighting interactions between Y8 in the N terminus and N482 (arrow) in the S6 helix.

and that the N-terminal residue Ala is acetylated, so the peptide lacks a free amino group capable of being charged, suggesting that voltage and external K^+ influence recovery from N-type inactivation through mechanisms that do not involve electrostatic repulsion.

Increasing the concentration of external K^+ also prevents slow C-type inactivation of the ion selectivity filter by stabilizing a conducting conformation[34,35,45]. N-type inactivation promotes slow C-type inactivation by blocking exposure of the filter to the high concentration of internal K^+ that the filter would otherwise see when the channel is open[46], but whether C-type inactivation might stabilize the N-type-inactivated state has not been explored. We therefore hypothesized that raising external K^+ and negative voltage might speed recovery from N-type inactivation by increasing occupancy of the selectivity filter, shifting channels from a C-type-inactivated state into a conducting state[34,35,45], and that the channel recovers more rapidly from N-type inactivation when the filter is conducting. To explore this possibility, we measured recovery from N-type inactivation with low and high external K^+ using longer test pulses to increase occupancy of the C-type-inactivated state in the WT channel (Fig. 4c) and observed that recovery from N-type inactivation in low external K^+ was slowed (Fig. 4c,d) compared with shorter test pulses (Fig. 1h and Extended Data Fig. 8c) and this effect

on recovery was largely reversed by increasing external K^+ (Fig. 4c,d and Extended Data Fig. 8d). We then introduced the T449V mutation in the selectivity filter, which stabilizes the conducting state of the filter, slowing C-type inactivation[34,35,45]. Although the kinetics for the onset of inactivation for T449V were similar to the WT channel (Fig. 4a,b), the rate of recovery from N-type inactivation was faster in T449V channels (Fig. 4c,e) than the WT channel (Fig. 4c,d) and less sensitive to raising external K^+ (Fig. 4e and Extended Data Fig. 8a–d). We concluded that C-type inactivation of the filter stabilizes the N-type-inactivated state of Shaker and that raising external K^+ speeds recovery from N-type inactivation by shifting channels from a C-type-inactivated state into a conducting state. Increasing external K^+ detectably speeds recovery from inactivation in the T449V mutant (Fig. 4c,e), suggesting that additional mechanisms are probably involved (see discussion).

## Interactions between inactivation particle and pore

Our structures showing engagement of an extended conformation of the N terminus with the internal pore are consistent with earlier studies suggesting that only one of the four N termini engages the internal pore[22] and that the N termini adopts an extended conformation[20,21,23,36,47,48].

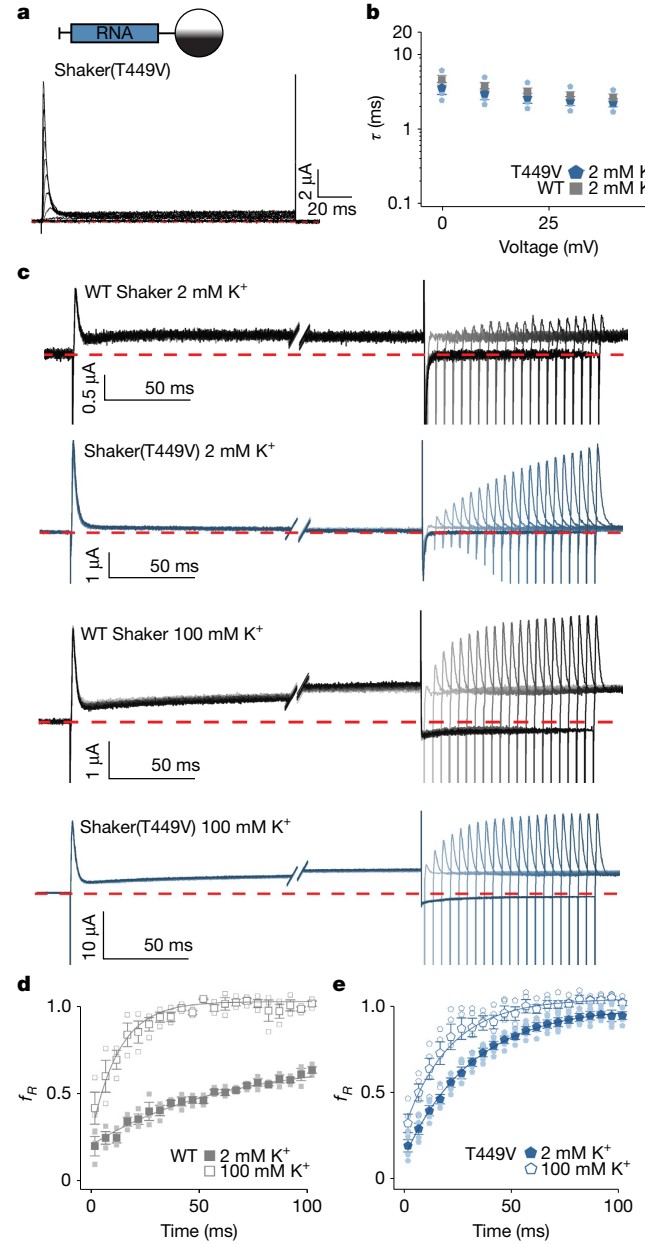

**Fig. 4 | Slow C-type inactivation regulates recovery from N-type inactivation.** **a**, Current traces for Shaker(T449V) expressed in oocytes by injecting RNA recorded in 2 mM external K⁺ from a holding voltage of −100 mV and step depolarizations from −100 mV to +80 mV (10-mV increments). **b**, Time constants of fast inactivation ($\tau$) plotted against test voltage in 2 mM external K⁺ for Shaker(T449V) compared with WT Shaker expressed by RNA injection. A single exponential fit to current traces like those in panel **a** was used to obtain values of $\tau$. Data for WT Shaker are from Fig. 1f and $n = 5$ cells in 2 independent experiments for Shaker(T449V). **c**, Family of current traces to measure recovery from inactivation for WT Shaker and Shaker(T449V) expressed by RNA injection. The holding voltage was −100 mV and initial step depolarizations were to +80 mV for 500 ms followed by varying amounts of time for recovery at −100 mV (initially 2 ms and increased in 5-ms intervals) before eliciting a second step depolarization to +80 mV. External K⁺ was either 2 mM or 100 mM as indicated. All traces shown are P/−4 subtracted, and the red dotted line denotes zero current. **d**, Fraction of current recovered at −100 mV ($f_R$) as a function of recovery time for WT Shaker expressed by RNA injection. For WT Shaker, $n = 3$ cells in 2 independent experiments for 2 mM external K⁺ and $n = 3$ in 2 independent experiments with 100 mM external K⁺. **e**, Fraction of current recovered at −100 mV ($f_R$) as a function of recovery time for T449V expressed by RNA injection. For T449V, $n = 5$ cells in 2 independent experiments for 2 mM external K⁺ and $n = 4$ in 2 independent experiments with 100 mM external K⁺. Error bars are s.e.m.

Our structures are also consistent with experiments showing that A391 within the S4–S5 linker helix is positioned near the inactivation particle in the N-type-inactivated state[49]. This position is within 4.5–6 Å of the density that we attributed to the N terminus in several of our structures (for example, Extended Data Fig. 6d,e). The direct interaction of the acetylated Ala with I470 is consistent with functional studies on mammalian K$_v$1 channels[14,23] and explains how RNA editing of this position to a smaller Val can diminish the extent of fast inactivation and hasten recovery from inactivation[14,48]. Indeed, functional experiments support direct hydrophobic interactions between position 2 and I470 in Shaker[48]. In our cryo-EM maps for AcA-EI Shaker, we saw interactions between V4 and both V474 and P475 in one subunit and A5 with those same two residues in the opposing subunit (Fig. 3g), consistent with functional studies showing that mutations at the equivalent to V474 in K$_v$1.4 channels diminish inactivation and are coupled with mutations of V3 in the inactivation particle of the β12 subunit[23].

To validate these structures, we interrogated two additional key regions in which structural relationships between the N terminus and the pore could be investigated using mutagenesis. First, the structures of GT Shaker and AcA-EI Shaker with free peptide show the sidechain of N482 in the S6 helix directed towards the N-terminal inactivation particle near Y8 at the innermost end of the pore (Fig. 3c,i). Although mutations at Y8 destabilize the inactivated state[20], their effects are modest, and in AcA-EI Shaker, the second N-terminal peptide in the chamber displaces Y8 so that it is no longer near N482 (Fig. 3f), suggesting that any interaction between Y8 and N482 is weak. However, the proximity of Y8 and N482 raises the possibility that stabilizing hydrophobic interactions could be generated by substitutions at N482. We therefore mutated N482 to Ala, Leu and Trp and observed that the mutants progressively slowed recovery from inactivation with introduction of larger hydrophobic sidechains (Fig. 5a–h and Extended Data Figs. 8e–h and 9d–f). N482W also appears to stabilize the open state because it shifted the voltage–activation relations to more negative voltages and profoundly slowed channel closure (Extended Data Fig. 9c,d), an alteration that would speed recovery from inactivation if occurring in isolation. That the N482W mutation so dramatically slows recovery from inactivation (Fig. 5e–h) therefore suggests that it probably stabilizes the inactivated state by interacting directly with hydrophobic residues (L7 and/or Y8) of the N terminus. By contrast, N480 is directed away from the inactivation particle in our structures and its mutation to Trp has little effect on inactivation (Extended Data Fig. 8i–l).

Second, we tested whether hydrophobic interactions between the hydrophobic triad of L7 in the N terminus, where mutations speed dissociation of the N terminus[20], with P475 and V478 in S6, stabilize the inactivated state. Although P475 may be important for inactivation because it is nearby L7 (Fig. 3b,e,h), mutations at P475 dramatically alter the functional properties of Shaker[40], probably owing to the unique influence of Pro residues on the helical structure of S6. However, mutation of V478 to Ala is well tolerated[40], so we tested whether this residue that forms the gate in the closed state[16,39,40] also contributes to stabilizing the inactivated state after the activation gate opens. When V478 was mutated to Ala, we observed slowing in the onset of inactivation, a reduction in the extent of inactivation and speeding of recovery from inactivation (Fig. 5i–l) without dramatically altering voltage-activation relation (Extended Data Fig. 9b), all of which are consistent with the intimate hydrophobic interactions between L7 and V478 and the likely importance of the hydrophobic triad involving L7, P475 and V478 in stabilizing the N-type-inactivated state.

## Discussion

The structural mechanism of fast N-type inactivation described here for Shaker can explain earlier functional studies on this K$_v$ channel, has important implications for the gating mechanisms of other K⁺ channels

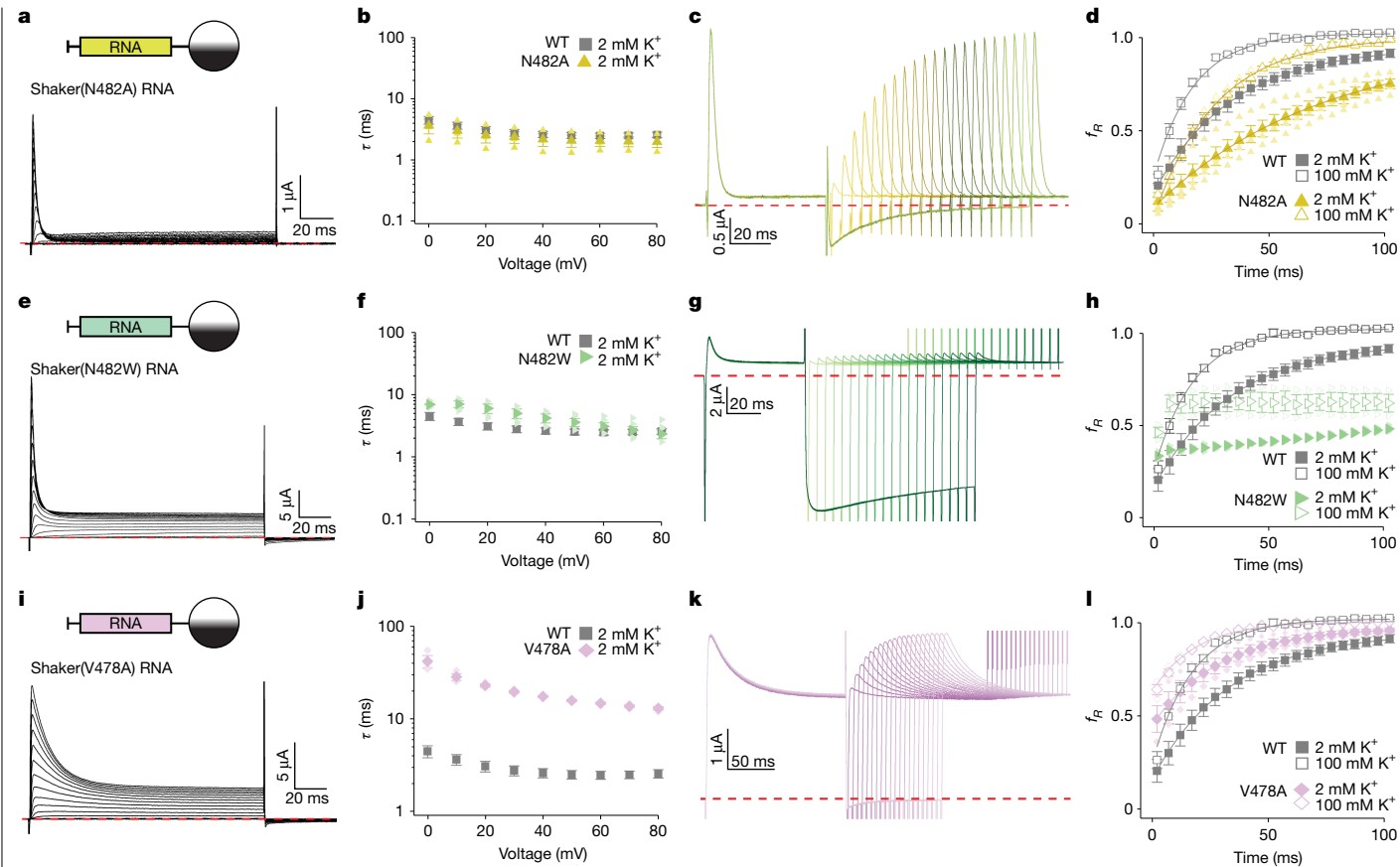

**Fig. 5 | Exploring key interactions of the N-terminal plug domain with the Shaker Kᵥ channel. a**, Currents recorded in 2 mM external K⁺ for N482A. The holding voltage was −100 mV, with 10-mV steps up to +80 mV. **b**, Time constants of fast inactivation ($\tau$) versus test voltage for Shaker(N482A) ($n$ = 3 cells in 2 independent experiments) compared with WT Shaker (from Fig. 1f). **c**, Current traces to measure recovery from inactivation for Shaker(N482A). The holding voltage was −100 mV, with initial 60-ms steps to +30 mV, followed by varying time for recovery at −100 mV (initially 2 ms and increased in 5-ms intervals) before eliciting a second step to +30 mV. External K⁺ was 100 mM. **d**, Fraction of current recovered at −100 mV ($f_R$) versus recovery time for N482A ($n$ = 4 cells in 3 independent experiments for 2 mM external K⁺ and $n$ = 3 in 2 independent experiments with 100 mM external K⁺). Data for WT Shaker are from Fig. 1h. **e**, Current traces for N482W using the same protocol as in panel **a**. **f**, Time constants of fast inactivation obtained as in panel **b** for N482W ($n$ = 6 cells in

3 independent experiments). **g**, Family of currents to measure recovery from inactivation for Shaker(N482W). The holding voltage was −120 mV, but otherwise the protocol and conditions are as in panel **c**. **h**, Fraction of current recovered at −100 mV versus recovery time for Shaker(N482W) ($n$ = 4 cells in 2 independent experiments for 2 mM external K⁺ and $n$ = 3 in 2 independent experiments with 100 mM external K⁺). **i**, Current traces for Shaker(V478A) using the same protocol as in panel **a**. **j**, Time constants of fast inactivation versus test voltage for Shaker (V478A) ($n$ = 3 cells in 2 independent experiments). **k**, Currents to measure recovery from inactivation for Shaker(V478A). The same protocol and conditions are as in panel **c** except initial steps were 150 ms. **l**, Fraction of current recovered at −100 mV versus recovery time for Shaker(V478A) ($n$ = 3 cells in 2 independent experiments for 2 mM external K⁺ and $n$ = 3 in 2 independent experiments with 100 mM external K⁺). P/−4 subtraction was used throughout, and the red dashed lines denote zero current. Error bars are s.e.m.

and underscores fundamental mechanistic differences with inactivation in Naᵥ channels. Our findings show that the N terminus forms a non-polar plug domain that engages with an open internal pore in an extended conformation, with the N-terminal acetylated Ala interacting intimately with the pore-lining I470 (Extended Data Fig. 10). The N terminus adopting an extended conformation is supported by mutagenesis studies[20,21,23,36,47], the temperature dependence for inactivation[20,21] and molecular dynamics simulations[36]. The interaction of the N-terminal acetylated Ala with I470 can explain how RNA editing at I470 regulates N-type inactivation, dramatically reducing the extent of inactivation in mammalian Kᵥ1 channels where the inactivation domain is provided by the N terminus of the β-subunit, and in the Shaker Kᵥ channel where the N terminus provides the inactivation domain[14,23,48]. Substitutions at I470 with less hydrophobic residues speed the rate of recovery from fast inactivation[14,48], consistent with a direct hydrophobic interaction between the N-terminal acetylated Ala and I470. This interaction is also supported by the finding that Cys substitution at A2 in the N terminus can form a disulfide bond with Cys substitution at I470 (ref. 48).

The pore-blocking mechanism that our structures support for the Shaker Kᵥ channel explains why these channels must first open before they can inactivate[24], why pore blockers such as TEA⁺ can compete with inactivation[25] and why channels can reopen and briefly conduct while recovering from inactivation[24] (Extended Data Fig. 10). Our structures also help to illuminate the mechanism by which raising the concentration of external K⁺ speeds recovery from inactivation, which was envisioned to occur by electrostatic interactions of K⁺ inside the pore with positively charged residues in the N terminus[24]. Instead, our structures show that only uncharged residues enter the pore during inactivation and that the amino group of the N-terminal Ala is acetylated and therefore cannot be positively charged. Our functional results show that C-type inactivation of the ion selectivity filter slows recovery from N-type inactivation (Fig. 4) and that raising external K⁺ speeds recovery from inactivation by drawing channels out of the C-type-inactivated state into the conducting state (Fig. 4). It will be fascinating to explore how the interplay between C-type inactivation of the filter and N-type inactivation of the internal pore influence electrical signalling in excitable cells as well as its underlying mechanism. We speculate that slowing

of recovery from N-type inactivation may be mediated by the network of interacting residues in the S6 helices that both line the permeation pathway and cradle the ion selectivity filter, perhaps conceptually related to the mechanism coupling pore opening and slow inactivation studied in the KcsA K[+] channel[50]. It is also interesting that raising external K[+] still detectably speeds recovery from N-type inactivation in the T449V mutant with disrupted C-type inactivation (Fig. 4c,e), suggesting that other mechanisms are involved. It will be interesting to explore whether van der Waals and molecular crowding or increased hydration of a hydrophobic cavity plugged by a hydrophobic N terminus may contribute to the influence of external K[+] and negative membrane voltage on recovery from inactivation.

One previously enigmatic feature of N-type inactivation in Shaker is that recovery follows a double-exponential time course[24,51,52]: a rapid phase of recovery occurring from channels that are open[24], which gets faster with hyperpolarization as K[+] ions are driven into the filter, and a slow phase occurring from channels that are closed, which gets slower with hyperpolarization[51]. Physical closure of the activation gate around a structured tethered blocker as originally envisioned in Na$_v$ channels[5,6] would be hard to imagine, but blockade of the internal pore by an extended conformation would leave enough space for the activation gate to squeeze the extended (and relatively dynamic) peptide chain to effectively stabilize it within the pore (Extended Data Fig. 10). Indeed, G6 within the N terminus is surrounded by P475 and V478 within the activation gate[16,39,40] (Fig. 3b,h, red asterisk), probably facilitating trapping of the N-terminal blocking particle as the gate closes.

In providing a structural foundation for how the N termini of K$_v$ channels or β-subunits mediate inactivation of these channels through a pore-blocking mechanism, our findings also provide a framework for clarifying why rapid inactivation in Na$_v$ channels is unlikely to occur through a similar mechanism[5,6]. In contrast to the findings discussed above for the Shaker K$_v$ channel, Na$_v$ channels can inactivate from closed states, they do not reopen during recovery from inactivation and negative membrane voltages actually promote recovery from inactivation[53], observations that are incompatible with inactivation through a pore-blocking mechanism. These key biophysical differences between inactivation in Na$_v$ and Shaker K$_v$ channels support recent proposals that Na$_v$ channels and some other K$_v$ channels inactivate by closure of the internal pore[10–13].

Inspection of the sequences of N-terminal inactivation domains from different K$_v$ channels, K$_v$ β-subunits and BK β-subunits reveals little conservation of sequence, except that residues near the N terminus tend to be non-polar[54,55], which our structure suggests is the likely region to enter into the pore during inactivation. Our unanticipated discovery that the N terminus of Shaker is post-translationally modified raises the intriguing possibility that related modifications of the N termini of other channels probably occur and it will be fascinating to elucidate the identity of the actual blocking particles in these K[+] channels and to examine how this might tune their mechanisms of fast inactivation. K$_v$ channels related to Shaker exhibiting fast N-type inactivation have recently been described in organisms within the oldest living branches of the animal kingdom[54,55], indicating that these channels and their mechanism of inactivation predate the evolutionary development of the nervous system.

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

## Methods

### Shaker K$_v$ channel expression

To produce the Shaker K$_v$ channel for cryo-EM, the full-length gene was cloned into the pEG vector in which eGFP was substituted with mVenus[56] to produce constructs with the mVenus tag on either the C or N terminus and a TEV protease site between mVenus and the channel. In addition, the full-length E12K/D13K Shaker K$_v$ channel was generated by introducing mutations into the WT construct with a C-terminal mVenus tag. These constructs were expressed in tsA201 cells (Sigma-Aldrich) using the previously published baculovirus-mammalian expression system with a few minor modifications[57]. In brief, P1 virus was generated by transfecting Sf9 cells (Thermo-Fisher; approximately 2.5 million cells on a T25 flask with a vent cap) with 10 µg of fresh Bacmid using Cellfectin (Thermo-Fisher). After 4–5 days of incubation in a humidified 27 °C incubator, the cell culture media were collected by centrifugation (at 4,000$g$ for 30 min), supplemented with 2% FBS and filtered through a 0.22-µm filter to harvest the P1 virus. To amplify the P1 virus, approximately 500 ml Sf9 cell cultures at a density of approximately 1 million cells per millilitre were infected with 1–200 µl of the virus and incubated in a 27 °C shaking incubator for 5 days. The cell culture media were then collected by centrifugation (at 5,000$g$ for 30 min), supplemented with 2% FBS and filtered through 0.22-µm filter to harvest P2 virus. The P2 virus was protected from light using aluminum foil and stored at 4 °C until use. To express the Shaker channels, tsA201 cells at approximately 1.5 million cells per millilitre in Freestyle medium with 1% FBS were transduced with 5% (v/v) P2 virus and incubated at 37 °C in a CO$_2$ incubator. To boost the protein expression, sodium butyrate (2 M stock in H$_2$O) was added to 10 mM at approximately 16 h post-transduction. The culture was continued at 30 °C in a CO$_2$ incubator for another 32 h, and the cells were harvested by centrifugation (at 5,000$g$ for 30 min) and frozen at −80 °C until use.

### Shaker K$_v$ channel purification

Before extraction of the Shaker channels from tsA201 cells, membrane fractionation was carried out using a hypotonic solution and ultracentrifugation. In our initial trials (such as those in Fig. 1b–d), cells were first resuspended in a hypotonic solution (20 mM Tris pH 7.5 and 10 mM KCl) supplemented with a cOmplete protease inhibitor cocktail tablet using a Dounce homogenizer, incubated at 4 °C for approximately 30 min, and centrifuged at 6,000$g$ for 10 min to remove cell debris. Once we realized there was considerable proteolysis of the N terminus, for EI Shaker, we increased protection from proteolysis by the addition of 1 µg ml$^{-1}$ pepstatin, 1 µg ml$^{-1}$ aprotinin, 1 µg ml$^{-1}$ leupeptin, 0.5 µg ml$^{-1}$ benzamidine, 0.1 µg ml$^{-1}$ soy trypsin inhibitor and 1.5 mM phenylmethylsulfonyl fluoride (PMSF) to the lysis buffer before homogenizing and replenished again after homogenizing. In all instances, the supernatant was ultracentrifuged for 1 h (at 195,000$g$); membranes were collected and stored at −80 °C until use. To purify Shaker K$_v$ channels, the fractionated membranes were resuspended in an extraction buffer (50 mM Tris pH 7.5, 150 mM KCl, 2 mM tris(2-carboxyethyl)phosphine hydrochloride (TCEP), 50 mM $n$-dodecyl-β-D-maltoside (DDM) and 5 mM cholesteryl hemisuccinate Tris salt (CHS) with the protease inhibitor mixture used above) and incubated for 1 h at 4 °C. The lysate was clarified by centrifugation (at 12,000$g$ for 10 min) and incubated with CoTALON resins (Takara) at 4 °C for 1 h. The mixture was transferred to an empty disposable column (Econo-Pac, Bio-Rad) and the resin was washed with 10 column volume of buffer A (50 mM Tris pH 7.5, 500 mM KCl, 1 mM DDM, 0.1 mM CHS and 0.1 mg ml$^{-1}$ porcine brain total lipid extract) with 20 mM imidazole before eluting bound proteins with buffer A with 250 mM imidazole. The eluate was concentrated using Amicon Ultra (Millipore; 100 kDa cut-off) to approximately 350–450 µl and loaded onto a Superose6 (Cytiva; 10/300) gel filtration column and separated with buffer (50 mM Tris pH 7.5, 150 mM KCl, 1 mM DDM and 0.1 mM CHS). All purification steps described above were carried out at 4 °C or on ice.

### Lipid nanodisc reconstitution of the Shaker K$_v$ channel

Lipid nanodisc reconstitution was performed following the previously published methods with minor modifications[33]. On the day of nanodisc reconstitution, the purified Shaker K$_v$ channel obtained from gel filtration in detergent was concentrated to approximately 1–3 mg ml$^{-1}$ and incubated with histidine-tagged MSP1E3D1 and 3:1:1 mixture of 1-palmitoyl-2-oleoyl-$sn$-glycero-3-phosphocholine (POPC), 1-palmitoyl-2-oleoyl-$sn$-glycero-3-phospho-(1′-rac-glycerol) (POPG) and 1-palmitoyl-2-oleoyl-$sn$-glycero-3-phosphoethanolamine (POPE) for 30 min at room temperature. The mixture was transferred to a tube with SM-2 Biobeads (approximately 30–50-fold of detergent, w/w) and incubated at room temperature for approximately 3 h with rotation in the presence of TEV protease (prepared in-house) and 2 mM TCEP to remove either N-terminal or C-terminal fusion protein including polyhistidine and the mVenus tag. The reconstituted protein was loaded onto a Superose6 column (10/300) and separated using 20 mM Tris pH 7.5, 4 mM KCl and 46 mM NaCl buffer at 4 °C. The success of nanodisc reconstitution was confirmed by collecting separated fractions and running SDS–PAGE to verify the presence of Shaker K$_v$ and MSP1E3D1 bands at a similar ratio. Typically, optimal reconstitution required the incubation of a 1:10:200 or 1:10:400 molar ratio of tetrameric Shaker K$_v$, MSP1E3D1 and the lipid mixture, respectively. The sample in the nanodisc was concentrated to 2.5–4 mg ml$^{-1}$.

### Mass spectrometry analysis

Protein samples for the Shaker K$_v$ channel were reconstituted in lipid nanodisc and run on SDS–PAGE. Excised bands for the Shaker proteins were reduced with 5 mM TCEP (Sigma-Aldrich), alkylated with 5 mM $N$-ethylmaleimide (NEM; Sigma-Aldrich) and digested with trypsin (Promega). Tryptic peptides were extracted then desalted before being injected into a nano-liquid chromatography with tandem mass spectrometry (nano-LC–MS/MS) system. For WT Shaker, LC–MS/MS data acquisition was carried out on an Orbitrap Ascend tribrid mass spectrometer (Thermo Scientific) with an EASY-Spray Ion Sources (Thermo Scientific) and coupled to a Vanquish Neo HPLC (Thermo Scientific). Of digests, 0.1–1 µg was loaded and desalted with an Acclaim PepMap 100 trapping column (75 µm × 2 cm; Thermo Scientific). Peptides were separated on an ES902 Easy-Spray column (75-µm inner diameter, 25 cm in length and 3 µm C18 beads; Thermo Scientific). The composition of mobile phase A (MPA) was 0.1% formic acid (Millipore-Sigma) in LC–MS grade water. The mobile phase B (MPB) was 0.1% formic acid in LC–MS grade acetonitrile (Millipore-Sigma). MPB was increased from 4% to 20% in 38 min. The flow rate was set at 300 nl min$^{-1}$. Mass spectrometers were operated in data-dependent mode. The resolution of the survey scan was set at 120,000. The $m/z$ range for the MS scan was 350–1,400. MS/MS scans were performed in the ion trap using higher-energy collisional dissociation (HCD) with the collision energy fixed at 30%. The minimum signal intensity required to trigger MS/MS scan was $1 \times 10^4$. MS1 scan was performed every 2 s. As many MS2 scans allowed were acquired within the MS1 scan cycle.

For GT Shaker samples and EI Shaker samples, an Ultimate 3000 HPLC (Thermo Scientific) and an Orbitrap Fusion Lumos Tribrid Mass Spectrometer (Thermo Scientific) were used for data acquisition. The LC–MS/MS method used was very similar to the one described above. The small differences included: MPB was increased from 5% to 22% in 39 min; the $m/z$ range for MS scan was 375–1,500; for GT Shaker, peptides were fragmented with an electron-transfer/higher-energy collision dissociation (EThcD) method.

Raw data were processed with Mascot Distiller and searched with Mascot Daemon software (Matrix Science). The search was performed against the house-built database containing the WT Shaker and EI Shaker sequences. The mass tolerances for precursor and fragment were set to 5 ppm and 0.5 Da, respectively. SemiTrypsin was used with up to three missed cleavages allowed. NEM on cysteines was set as fixed

modification. Variable modifications included oxidation (M), Met-loss (protein N-term), Met loss + acetyl (protein N-term) and acetyl (N-term). The search results were filtered by a false discovery rate of 1% at the protein level. Peptides detected by database search were manually curated.

## Reconstitution of Shaker $K_v$ channels into liposomes for injection into oocytes

On the day of reconstitution, the Shaker $K_v$ channel purified by Superose6 in detergent was concentrated to 1–3 mg ml$^{-1}$ and incubated for 30 min at room temperature with a solution containing 20 mg ml$^{-1}$ of a 3:1:1 mixture of POPC, POPG and POPE where the mass ratio of protein to lipid was 1:10. The mixture was transferred to a tube with SM-2 Biobeads (approximately 30–50-fold of detergent, w/w) and incubated at room temperature for approximately 3 h with rotation either in the absence or presence of TEV protease (prepared in-house) with 2 mM TCEP to remove either N-terminal or C-terminal fusion protein including polyhistidine and mVenus tags. Biobeads were allowed to settle, and the resulting solution used for injection into oocytes. The final concentration of protein in liposome was 1 mg ml$^{-1}$.

## Cryo-EM sample preparation and data acquisition

For C-terminal mVenus-tagged Shaker, concentrated sample in nanodiscs (3 μl), with or without the addition of 1.5 mM fluorinated Fos-choline-8 (Anatrace) were applied to glow-discharged Quantifoil grids (R1.2/1.3 Cu 300 mesh). For GT Shaker, 3 μl sample in nanodiscs was applied to glow-discharged Quantifoil grids (R1.2/1.3 Cu 300 mesh). For the full-length E12K/D13K AcA-EI Shaker construct, the sample was applied to glow-discharged Ultrafoil grids (R1.2/1.3 Au 300 mesh). For the full-length E12K/D13K AcA-EI Shaker sample with 1 mM free N-terminal peptide, the peptide was incubated with protein for 30 min before grid preparation. Acetylated free N-terminal peptide (Ac-AAVAGLYGLGKKRQHRKKQ; molecular weight 2,151 Da) was synthesized by GenScript. After incubation, 3 μl sample was applied to glow-discharged Ultrafoil grids (R1.2/1.3 Au 300 mesh). The grids were blotted for 2.5 s, with a blot force of 4, at 100% humidity at 16 °C using a FEI Vitrobot Mark IV (Thermo Fisher), followed by plunging into liquid ethane cooled by liquid nitrogen.

Images were acquired using an FEI Titan Krios equipped with a Gatan LS image energy filter (slit width of 20 eV) operating at 300 kV. A Gatan K3 Summit direct electron detector was used to record movies in super-resolution mode with a nominal magnification of ×105,000, resulting in a calibrated pixel size of 0.415 Å per pixel. The typical defocus values ranged from −0.5 to −2.0 μm. Exposures of 2 s were dose fractionated into 40 frames, resulting in a total dose of 52 e$^-$ Å$^{-2}$. Movies were recorded using the automated acquisition program SerialEM[58].

## Image processing

All processing was completed in RELION[59] and cryoSPARC[60]. In general, the beam-induced sample motion between frames of each dose-fractionated micrograph was corrected and binned by 2 using Motion-Cor2 (ref. 61) or Patch Motion Correction, and contrast transfer function (CTF) estimation was performed using CTFFIND4 (ref. 62) or Patch CTF estimation. Micrographs were selected and those with outliers removed based on defocus value and astigmatism, as well as low resolution (more than 5 Å) reported by CTF estimation. The initial set of particles from subset micrographs were picked using Blob picker or Gautomatch (https://www2.mrc-lmb.cam.ac.uk/research/locally-developed-software/zhang-software/#gauto), followed by reference-free 2D classification. The good classes were then used as template to pick particles from all selected micrographs using a different program (including Gautomatch, Topaz pick or Template Picker). The best particles were selected iteratively by selecting the 2D class averages and 3D reconstructions (using C1 or C4 symmetry) that had interpretable structural features. After performing non-uniform refinement in either C1 or C4 symmetry, the unsharpened map of the

C-terminal mVenus-tagged Shaker exhibited significantly weaker density in the internal pore than GT Shaker. Therefore, we used GT Shaker as the model for subsequent data processing.

For GT Shaker, the best particles were aligned using 3D autorefine in C4 symmetry in RELION. To further classify the particles, the particles were expanded from C4 to C1 symmetry. These particles were submitted to 3D classification for 10 classes without alignment. A cylindrical mask covering the pore region of two neighbouring monomers and the corresponding chamber between the transmembrane and T1 domains was used for 3D classification. Among 10 classes, one good class clearly showed L-shape density extending from the chamber to the internal pore, and this L-shape density was used to generate a mask for further classification (Extended Data Fig. 3).

To obtain a high-resolution map of the internal pore and chamber, the best particles were imported to cryoSPARC and subjected to NU Refinement with applied C4 symmetry. The 1.4 million aligned particles were submitted to 3D classification using the principal component analysis (PCA) mode in C1 symmetry with L-shape mask (10 classes, a filter resolution of 3 Å, online expectation-maximization [O-EM] batch size of 8,000 and O-EM learning rate init of 0.6–0.8) for 3 rounds. Four classes (1.1 million particles), exhibiting L-shape density within a 90° rotation from one to the next, were applied to Align 3D Maps. To separate different conformations of the N-terminal peptide within the internal pore (for example, Extended Data Fig. 6c), the aligned particles (1.1 million) were then further classified by changing the O-EM learning rate init to 0.0001 and the F-EM iters to 100. The best class (141,800) was selected and subjected to final local refinement with full-length protein mask in C1 symmetry. The final reconstruction was reported at 2.94 Å (Extended Data Fig. 3).

The other classes (424,000) showed density occupying in the chamber without extending into the internal pore. Further classification was applied for these particles, which generated two different classes, following local refinement in C1 symmetry. The resolutions of final constructs were reported at 2.79 Å and 3.01 Å (Extended Data Fig. 3).

During data processing, we noticed that the resolution of the T1 domain is much lower than the transmembrane domain. To get a high-resolution map of the T1 domain, the density of the T1 domain was obtained by subtracting the transmembrane region from the full-length protein. The subtracted particles were then submitted to 3D classification using PCA mode with a T1 domain mask in C1 symmetry by changing the filter resolution to 6 Å. All the classes were aligned using Align 3D Maps and followed by local refinement in C4 symmetry. The final reconstruction for the T1 domain was reported at 2.31 Å (Extended Data Fig. 3).

The other datasets for Shaker with a C-terminal mVenus tag (Supplementary Fig. 1), and the AcA-EI Shaker construct in the absence (Supplementary Fig. 2) or presence (Supplementary Fig. 3) of additional N-terminal peptide were processed using a method similar to GT Shaker.

## Model building and structure refinement

Model building was first carried out by manually fitting the transmembrane domain of Shaker (Protein Data Bank ID: 8TEO) and the T1 domain generated by AlphaFold3 (ref. 63) into the electron microscopy density map using UCSF Chimera[64]. The model was then manually built in Coot[65] and refined using real_space_refine in PHENIX[66] with secondary structure and geometry restraints. The final model was evaluated by comprehensive validation in PHENIX. Structural figures were generated using PyMOL (https://pymol.org/2/support.html), UCSF Chimera[64] and UCSF ChimeraX[67].

## Electrophysiological recordings

For electrophysiological recordings, the full-length Shaker $K_v$ channel cDNAs were cloned into the pGEM-HE vector[68]. Mutagenesis was performed by the QuikChange Lightning Kit (Agilent) using the full-length

channel unless otherwise indicated. The DNA sequence of all constructs and mutants was confirmed by automated DNA sequencing. cRNA was synthesized using the T7 polymerase (mMessage mMachine Kit, Ambion) after linearizing with Nhe-I (NEB).

Oocytes (stage V–VI) from female *Xenopus laevis* frogs (approximately 1–2 years of age from *Xenopus* I) were removed surgically and incubated for 1 h at 19 °C in a solution containing: NaCl (82.5 mM), KCl (2.5 mM), MgCl$_2$ (1 mM), HEPES (5 mM), pH 7.6, with NaOH and collagenase type II (2 mg ml$^{-1}$; Worthington Biochemical). The animal care and experimental procedures were performed in accordance with the Guide for the Care and Use of Laboratory Animals and were approved by the Institutional Animal Care and Use Committee of the National Institute of Neurological Disorders and Stroke (animal protocol number 1253). Defolliculated oocytes were injected with cRNA or liposomes containing reconstituted Shaker K$_v$ channels and incubated at 16 °C in a solution containing: NaCl (96 mM), KCl (2 mM), MgCl$_2$ (1 mM), CaCl$_2$ (1.8 mM), HEPES (5 mM), pH 7.6 (with NaOH), and gentamicin (50 mg ml$^{-1}$; GIBCO-BRL) for 24–72 h before electrophysiological recording. Oocyte membrane voltage was controlled by an OC-725C oocyte clamp (Warner Instruments) and controlled using pClamp (10.7). Data were filtered at 1 kHz (8-pole Bessel) and digitized at 5–10 kHz. Microelectrode resistances ranged from 0.2 to 0.6 MΩ when filled with 3 M KCl. Oocytes were studied in 150 µl recording chambers that were perfused continuously with an extracellular solution containing: NaCl (98 mM), KCl (2 mM), MgCl$_2$ (1 mM), CaCl$_2$ (0.3 mM) and HEPES (5 mM), pH 7.6, with NaOH. When other external K$^+$ concentrations were used, NaCl was replaced with KCl. Most experiments were undertaken in lower external K$^+$ to approximate physiological conditions, and elevated external K$^+$ was used in some experiments where inward tail currents were measured to compare the gating properties of different mutants. All experiments were done using a continuous flowing external solution and were carried out at room temperature (22 °C). Leak and background conductances were subtracted for tail current measurements by arithmetically deducting the end of the tail pulse of each analysed trace. In most instances, K$_v$ channel currents shown are non-subtracted, but where indicated, a P/−4 leak subtraction protocol was used.

The Boltzmann equation was fit to G–V relations to obtain the $V_{1/2}$ and $z$ values according to

$$\frac{I}{I_{max}} = \left( 1 + e^{-zF(V - V_{1/2})/RT} \right)$$

where $z$ is the equivalent charge, $V_{1/2}$ is the half-activation voltage, $F$ is Faraday's constant, $R$ is the gas constant and $T$ is temperature in Kelvin. Time constants of inactivation were obtained by fitting a single or double exponential function to the decay of currents using the following equation:

$$f(t) = \sum_{i=0}^{n} A_i e^{-t/\tau_i} + C$$

where $A$ is the amplitude and $\tau$ is the time constant. All analyses of electrophysiological data were done using Origin 2023b.

## Sample size
Statistical methods were not used to determine sample size. Sample size for cryo-EM studies was determined by availability of microscope time and to ensure sufficient resolution for model building. Sample size for electrophysiological studies was determined empirically by comparing individual measurements with population data obtained under differing conditions until convincing differences or lack thereof were evident. For all electrophysiological experiments, $n$ values represent the number of oocytes studied between 2 and 10 different frogs (indicated as independent experiments).

## Data exclusions
For electrophysiological experiments, exploratory experiments were undertaken with varying ionic conditions and voltage-clamp protocols to define ideal conditions for measurements reported in this study. Although these preliminary experiments are consistent with the results that we report, they were not included in our analysis due to varying experimental conditions (for example, solution composition and voltage protocols). Once ideal conditions were identified, electrophysiological data were collected for control and mutant constructs until convincing trends in population datasets were obtained. Individual cells were also excluded if cells exhibited excessive initial leak currents at the holding voltage (more than 0.5 µA), if currents arising from expressed channels were too small (less than 0.5 µA), making it difficult to distinguish the activity of expressed channels from endogenous channels, or if currents arising from expressed channels were too large, resulting in substantial voltage errors or changes in the concentration of ions in either intracellular or extracellular solutions.

## Randomization and blinding
Randomization and blinding were not used in this study. The effects of different conditions or mutations on Shaker K$_v$ channels heterologously expressed in individual cells was either unambiguously robust or clearly indistinguishable from control conditions.

## Reporting summary
Further information on research design is available in the Nature Portfolio Reporting Summary linked to this article.

## Data availability
All data needed to evaluate the conclusions in the paper are present in the paper and/or the Supplementary Information. Models of Shaker have been deposited in the Protein Data Bank with accession codes 9NES for the Shaker transmembrane domain in C4, 9NEU for the Shaker T1 domain in C4, 9NEI for GT Shaker class A, 9NEG for AcA-EI Shaker class C, 9NEC for Aca-EI Shaker + free peptide conformation A and 9NED for Aca-EI Shaker + free peptide conformation B. Maps of Shaker have been deposited in the Electron Microscopy Data Bank under accession codes EMD-49333 for the Shaker transmembrane domain in C4, EMD-49336 for the Shaker T1 domain in C4, EMD-49331 for the GT Shaker transmembrane domain in C4, EMD-49312 for the GT Shaker T1 domain in C4, EMD-49311 for GT Shaker class A in C1, EMD-49337 for GT Shaker class B in C1, EMD-49338 for GT Shaker class C in C1, EMD-49356 for AcA-EI Shaker class A in C1, EMD-49365 for AcA-EI Shaker class B in C1, EMD-49308 for AcA-EI Shaker class C in C1 and EMD-49305 Aca-EI Shaker + free peptide in C1. Additional datasets used in this study include Protein Data Bank accession code 8TEO.

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

**Acknowledgements** We thank A. Jara-Oseguera, M. Holmgren, M. Mayer and members of the Swartz laboratory for helpful discussion; and H. Wang and U. Baxa in the US National Institutes of Health (NIH) Multi-Institute Cryo-EM Facility (MICEF) and Z. Yu in the National Institute of Neurological Disorders and Stroke (NINDS) Cryo-EM Core Facility for assistance in acquiring cryo-EM data. This work was supported by the Division of Intramural Research of the NINDS (NIH) to K.J.S. (NS002945) and utilized the National Institute of Diabetes and Digestive and Kidney Diseases and NINDS Cryo-EM Core Facilities, the NIH MICEF and computational resources of the NIH High-Performance Computing Biowulf cluster (http://hpc.nih.gov).

**Author contributions** X.-F.T., A.I.F.-M. and K.J.S. conceptualized the study. X.-F.T., A.I.F.-M., Y.L., T.-H.C. and K.J.S. provided the methodology. X.-F.T., A.I.F.-M., Y.L., T.-H.C. and K.J.S. undertook the investigation. X.-F.T., A.I.F.-M., Y.L. and K.J.S. performed the visualization. K.J.S. acquired funding. X.-F.T., A.I.F.-M. and K.J.S. provided project administration. K.J.S. supervised the study. X.-F.T., A.I.F.-M. and K.J.S. wrote the original draft of the manuscript. X.-F.T., A.I.F.-M., Y.L. and K.J.S. reviewed and edited the manuscript.

**Competing interests** The authors declare no competing interests.

**Additional information**
**Correspondence and requests for materials** should be addressed to Xiao-Feng Tan, Ana I. Fernández-Mariño or Kenton J. Swartz.

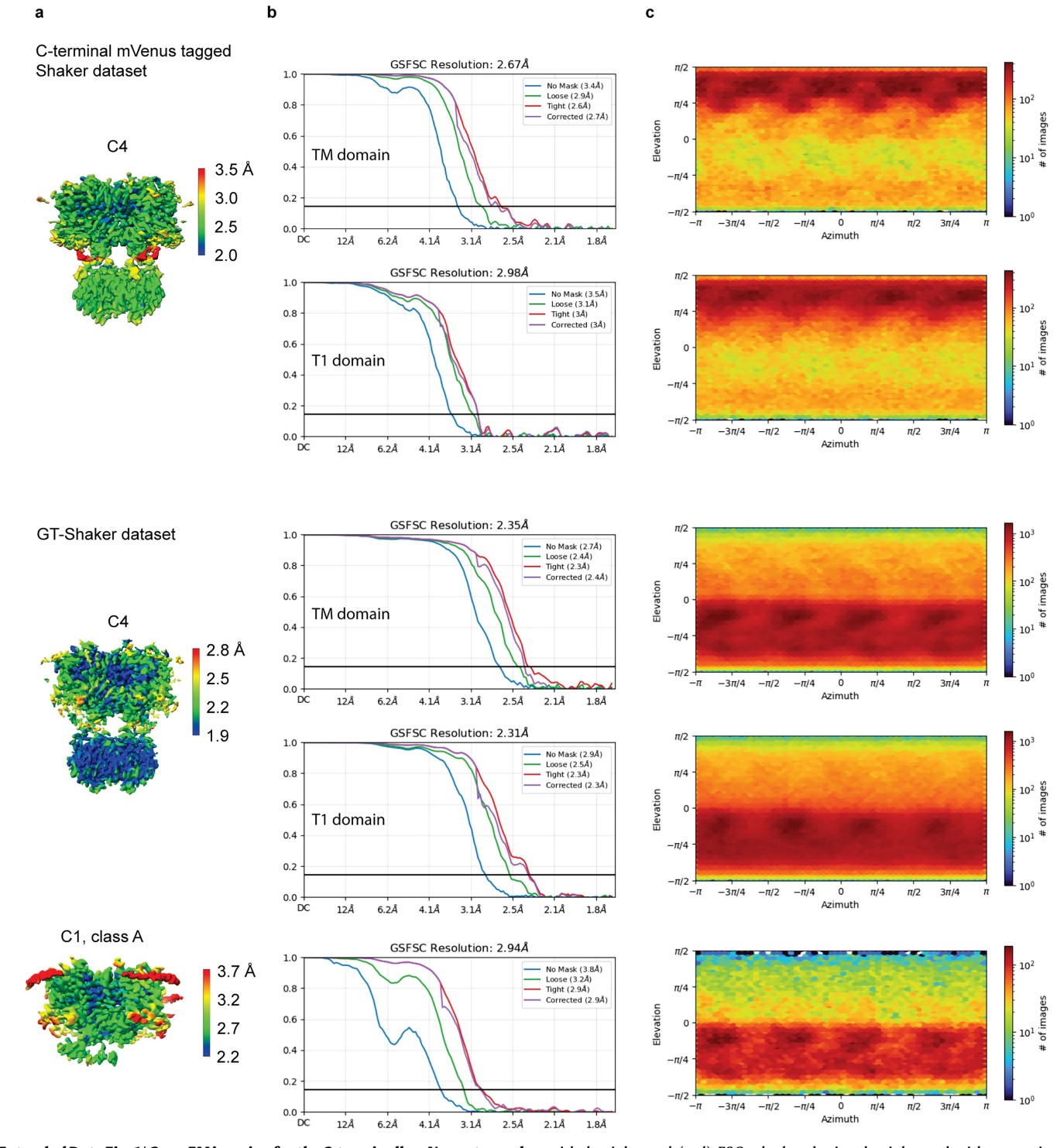

**Extended Data Fig. 1 | Cryo-EM imaging for the C-terminally mVenus tagged Shaker Kv channel and GT-Shaker constructs. a**) Local resolution maps for each structure. **b**) Fourier Shell Correlation (FSC) curves: FSC calculated without mask (blue); FSC calculated with a soft solvent mask (green); FSC calculated with the tight mask (red); FSC calculated using the tight mask with correction by noise substitution (purple). **c**) Direction distribution plots of the 3D reconstruction illustrating the distribution of particles in different orientations.

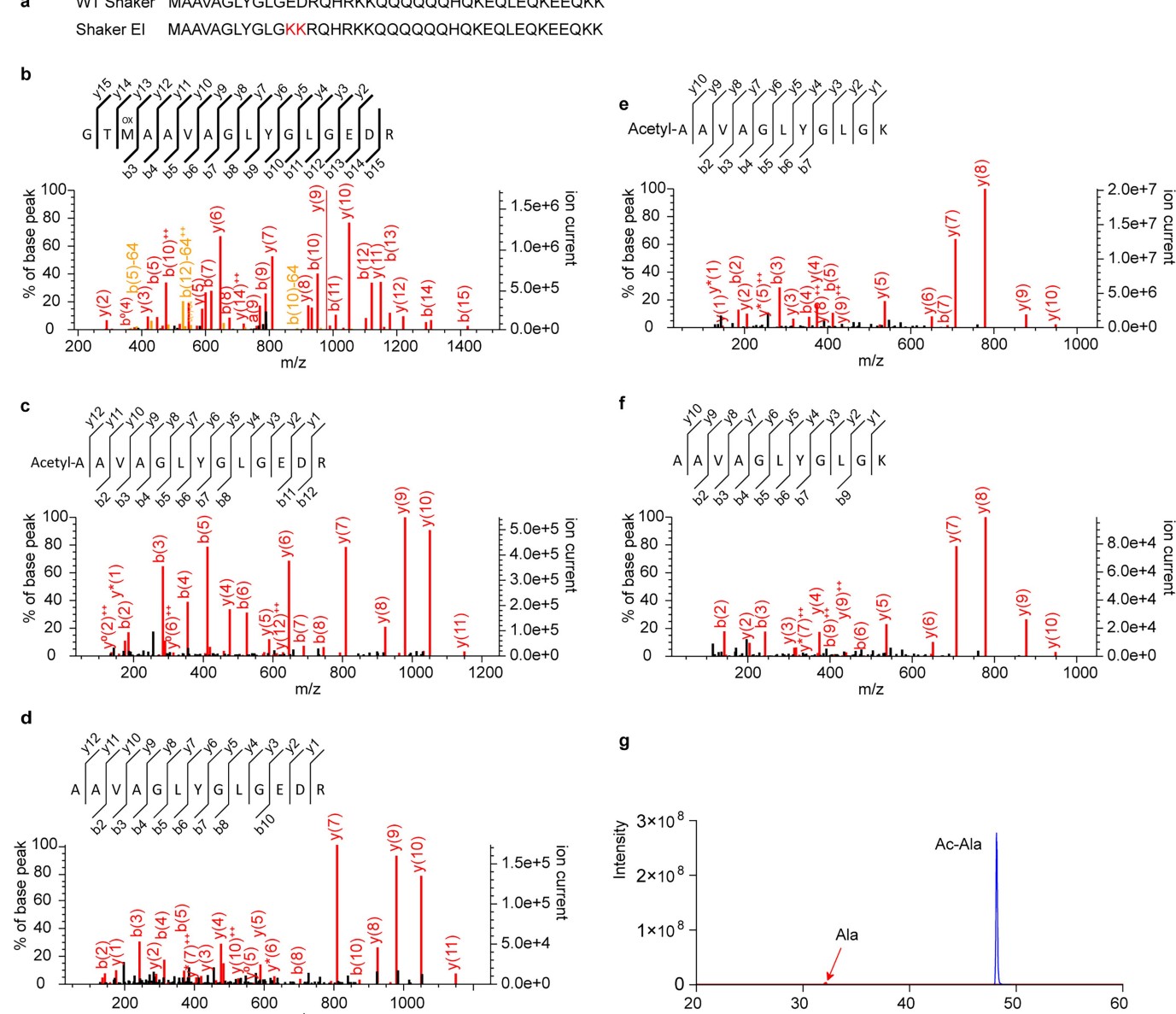

**a**

WT Shaker  MAAVAGLYGLGEDRQHRKKQQQQQQHQKEQLEQKEEQKK

Shaker EI  MAAVAGLYGLG**KK**RQHRKKQQQQQQHQKEQLEQKEEQKK

**Extended Data Fig. 2 | Mass spectrometry of the full-length Shaker Kv channel. a**) N-terminal sequences of WT-Shaker and EI-Shaker. **b**) MS/MS spectrum of an N-terminal peptide containing an additional Gly and Thr resulting from cleavage of the N-terminally mVenus-tagged Shaker Kv channel with TEV protease. **c**) MS/MS spectrum of an N-terminal peptide containing deletion of the N-terminal Met and acylation of Ala2 for the C-terminally mVenus-tagged Shaker Kv channel. **d**) MS/MS spectrum of an N-terminal peptide containing deletion of the N-terminal Met for the C-terminally

mVenus-tagged Shaker Kv channel. **e**) MS/MS spectrum of an N-terminal peptide containing deletion of the N-terminal Met and acylation of Ala2 for the C-terminally mVenus-tagged EI-Shaker Kv channel. **f**) MS/MS spectrum of an N-terminal peptide containing deletion of the N-terminal Met for the C-terminally mVenus-tagged EI-Shaker Kv channel. **g**) Extracted ion chromatogram (XIC) of (Ac)AAVAGLYGLGK (Ac-Ala, labeled in blue) and AAVAGLYGLGK (Ala, labeled in red).

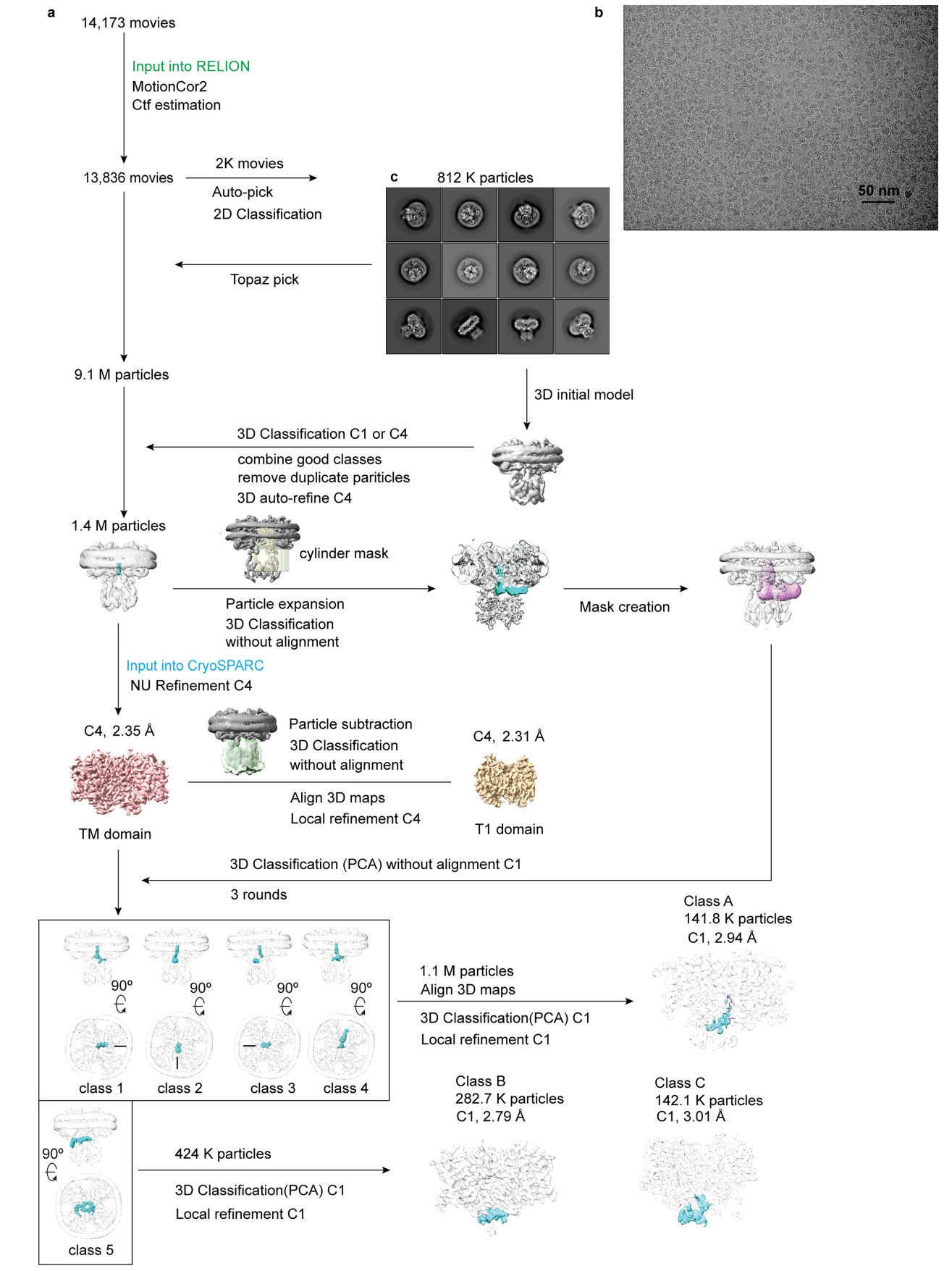

**Extended Data Fig. 3 | Data processing workflow for the cryo-EM structure of the GT-Shaker Kv channel. a**) Cyro-EM data processing pipeline for the GT-Shaker Kv channel. **b**) Representative micrograph from the 14,173 movies collected. About 2.4% of micrographs were discarded and at least 50% were of comparable quality to the representative one shown. **c**) 2D class averages of particles showing different orientations.

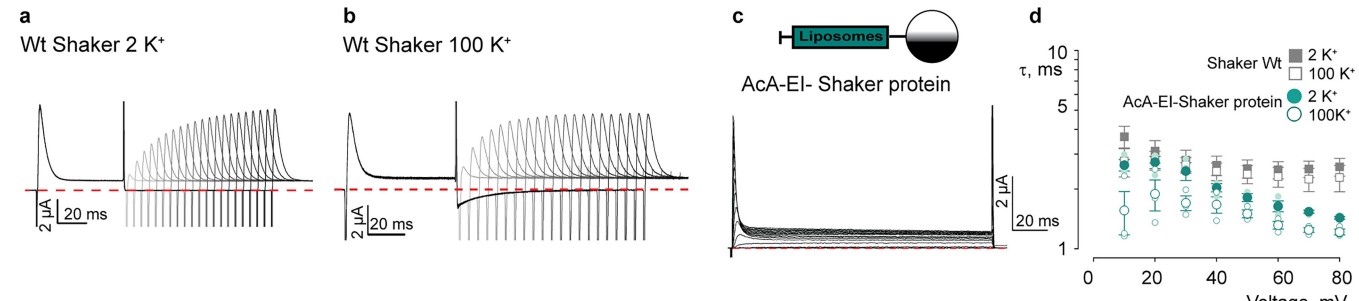

**a**
Wt Shaker 2 K$^+$

**b**
Wt Shaker 100 K$^+$

**c**
AcA-EI- Shaker protein

**d**

**Extended Data Fig. 4 | Recovery from inactivation for WT Shaker and inactivation of purified AcA-EI-Shaker. a,b**) Family of current traces to measure recovery from inactivation for WT-Shaker expressed by RNA injection with 2 mM (a) or 100 mM (b) external K$^+$. Holding voltage was −100 mV and initial step depolarizations were to +30 mV for 60 ms followed by varying amounts of time for recovery at −100 mV (initially 2 ms and increased in 5 ms intervals) before eliciting a second step depolarization to +30 mV. Traces shown are P/−4 subtracted and red dotted line denotes zero current. Population data for recovery is shown in Fig. 1h. **c**) Current traces for AcA-EI-Shaker expressed in oocytes by injecting liposomes containing reconstituted purified proteins expressed in HEK cells. Oocytes were recorded in 2 mM external K$^+$ from a holding voltage of −100 mV and step depolarizations from −100 mV to +80 mV (10 mV increments). Traces shown are P/−4 subtracted and red dotted line denotes zero current. **d**) Time constants of fast inactivation (τ) plotted against test voltage in WT-Shaker expressed by RNA injection compared to AcA-EI-Shaker expressed by proteoliposome injection. A single exponential fit to current traces like those in panel a was used to obtain values of τ. Data for WT-Shaker are from Fig. 2e and for AcA-EI-Shaker n = 3 cells in 2 independent experiments for 2 mM external K$^+$ and n = 3 cells in 2 independent experiments for 100 mM external K$^+$. Error bars are S.E.M.

**a**

### AcA-El Shaker

C1, class C

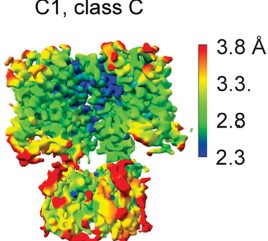

**b**

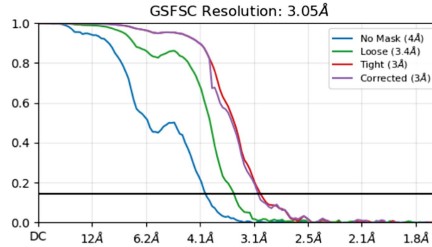

**c**

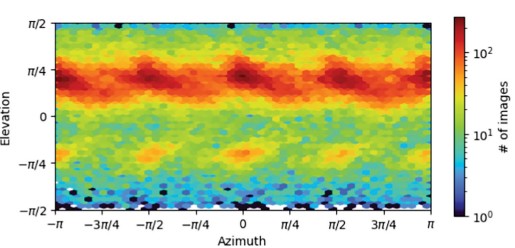

### AcA-El Shaker Kv plus free peptide

C1

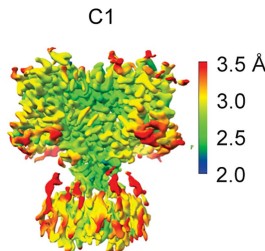

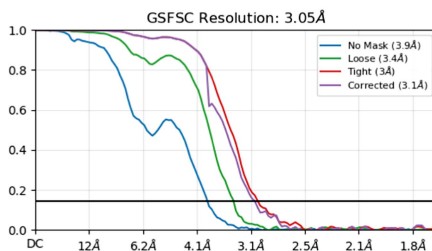

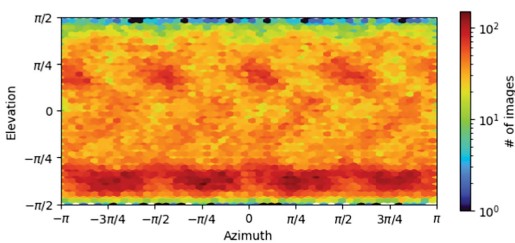

**Extended Data Fig. 5 | Cryo-EM imaging for full-length Shaker Kv channel with enhanced inactivation mutations. a**) Local resolution maps for each structure. **b**) Fourier Shell Correlation (FSC) curves: FSC calculated without mask (blue); FSC calculated with a soft solvent mask (green); FSC calculated with the tight mask (red); FSC calculated using the tight mask with correction by noise substitution (purple). **c**) Direction distribution plots of the 3D reconstruction illustrating the distribution of particles in different orientations.

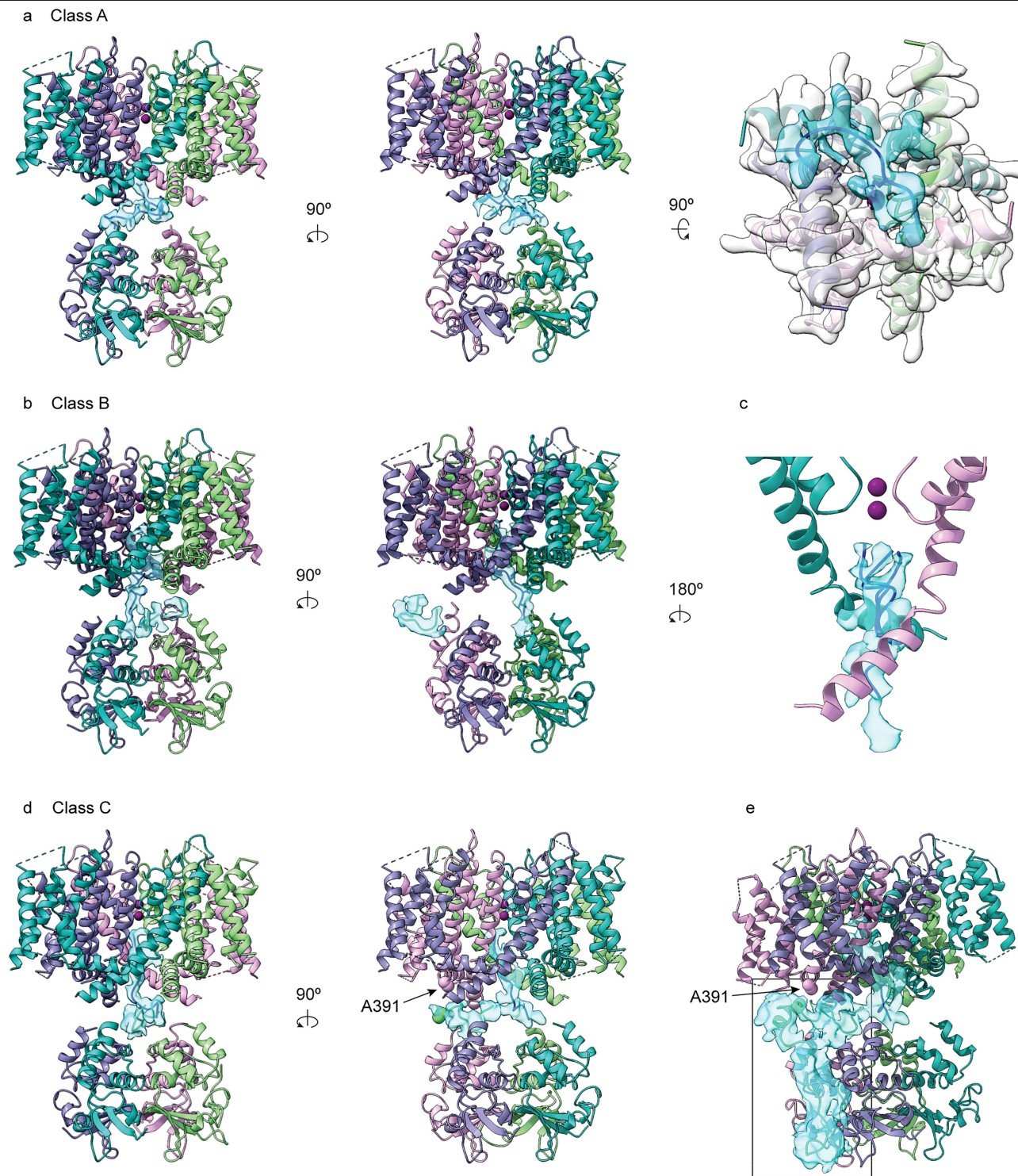

**Extended Data Fig. 6 | Cryo-EM structures for the full-length Shaker Kv channel with enhanced inactivation mutations. a**) Structural model for class A of the AcA-EI-Shaker Kv channel with cryo-EM density for the N-terminus shown in light blue from three rotational perspectives. **b**) Structural model for class B of the AcA-EI-Shaker Kv channel with cryo-EM density for the N-terminus shown in light blue from two rotational perspectives. **c**) Magnified N-terminal cryo-EM density for class B AcA-EI-Shaker, likely representing the average density for three conformations of the N-terminal peptide. **d**) Structural model

for class C of the AcA-EI-Shaker Kv channel with cryo-EM density for the N-terminus shown in light blue from two rotational perspectives. In the second perspective, A391 in the S4-S5 linker is within 4.5–6 Å of the N-terminal cryo-EM density. **e**) Additional low-resolution cryo-EM density for a second N-terminal peptide in AcA-EI-Shaker class C (black box) connecting density within the chamber to the T1 domain. A391 in the S4-S5 linker is within 4.5–6 Å of the N-terminal cryo-EM density.

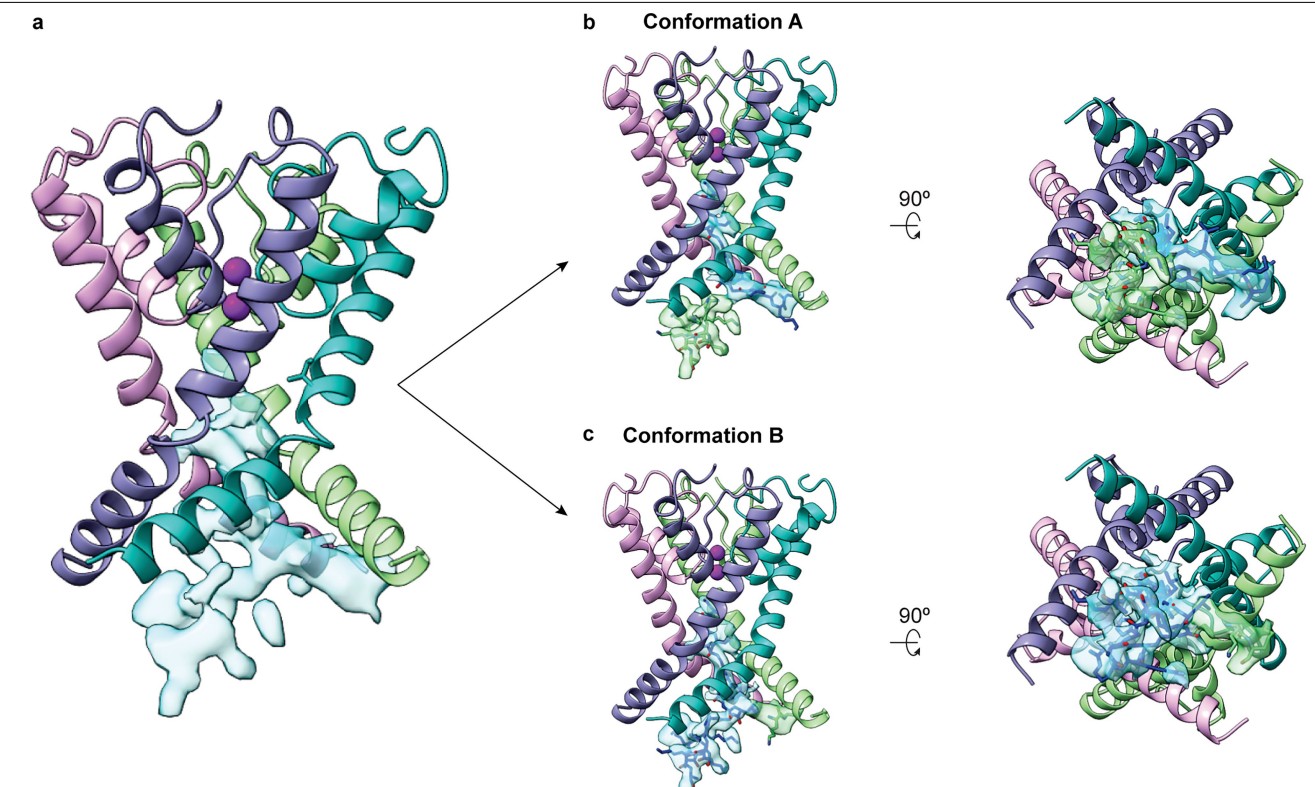

**a**

**b** **Conformation A**

90°

**c** **Conformation B**

90°

**Extended Data Fig. 7 | Cryo-EM structures for the full-length Shaker Kv channel with enhanced inactivation mutations in the presence of free N-terminal peptide. a)** Structural model of the pore domain S6 helices for AcA-EI-Shaker Kv channel in the presence of additional N-terminal peptide with cryo-EM density for the N-terminus shown in light blue. **b)** Model of conformation A where two N-terminal peptides (blue and green) were modeled into the cryo-EM density. **c)** Model of conformation B where two N-terminal peptides (blue and green) were modeled into the cryo-EM density.

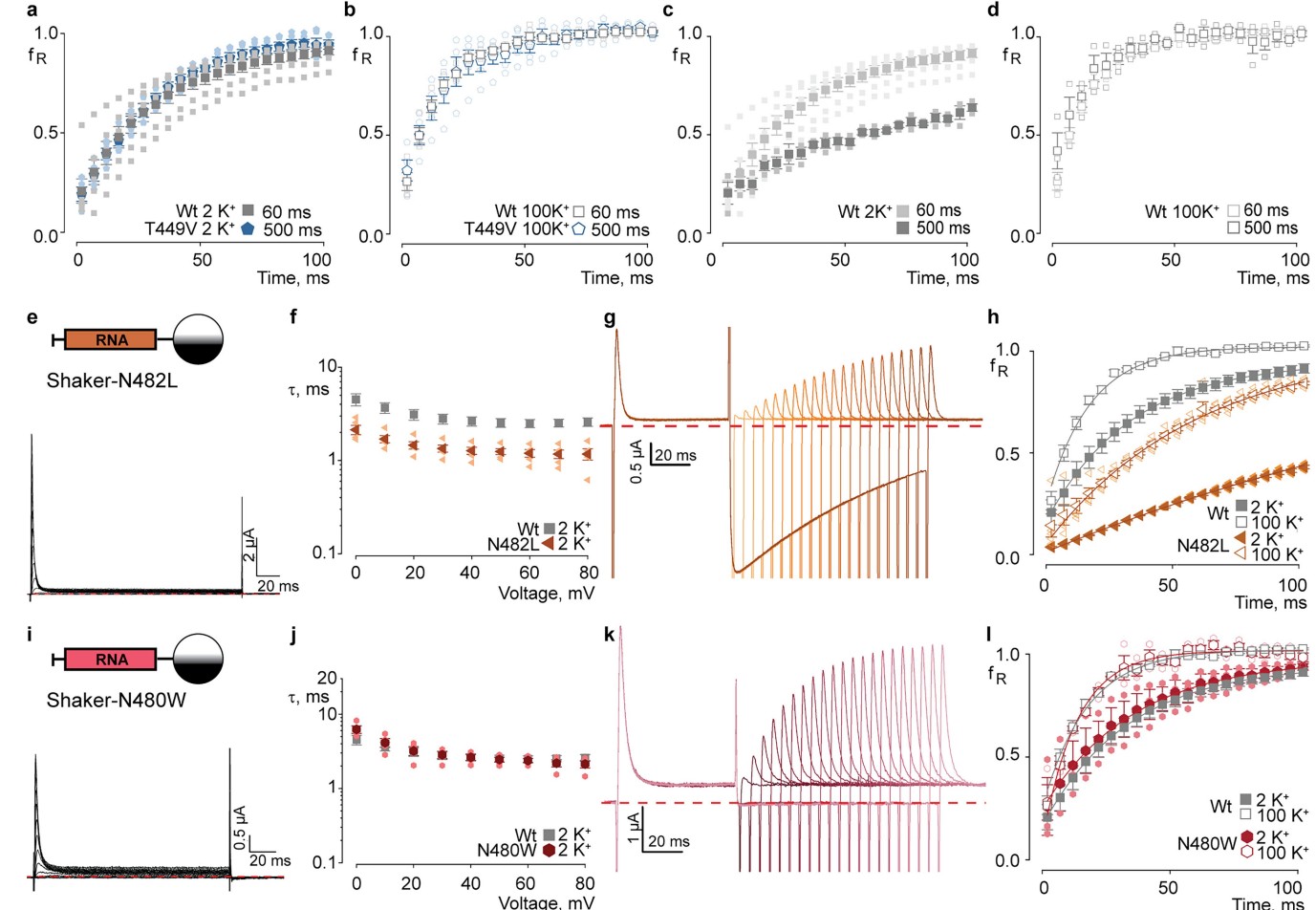

**Extended Data Fig. 8 | Inactivation and recovery from inactivation for mutants of Shaker. a)** Fraction of current recovered at −100 mV ($f_R$) as a function of recovery time for WT-Shaker with a 60 ms inactivating depolarization to +30 mV compared to T449V with a 500 ms inactivating depolarization to +80 mV, both in 2 mM external K⁺. Data for WT-Shaker are from Fig. 1h and that for T449V are from Fig. 4e. **b)** Fraction of current recovered at −100 mV ($f_R$) as a function of recovery time for WT-Shaker with a 60 ms inactivating depolarization to +30 mV compared to T449V with a 500 ms inactivating depolarization to +80 mV, both in 100 mM external K⁺. Data for WT-Shaker are from Fig. 1h and that for T449V are from Fig. 4e. **c)** Fraction of current recovered at −100 mV ($f_R$) as a function of recovery time for WT-Shaker with a 60 ms inactivating depolarization to +30 mV compared to a 500 ms inactivating depolarization to +80 mV, both in 2 mM external K⁺. Data for the 60 ms inactivating pulse are from Fig. 1h and that for the 500 ms inactivating pulse are from Fig. 4d. **d)** Fraction of current recovered at −100 mV ($f_R$) as a function of recovery time for WT-Shaker with a 60 ms inactivating depolarization to +30 mV compared to a 500 ms inactivating depolarization to +80 mV, both in 100 mM external K⁺. Data for the 60 ms inactivating pulse are from Fig. 1h and that for the 500 ms inactivating pulse are from Fig. 4d. **e)** Current traces for Shaker N482L expressed in oocytes by injecting RNA recorded in 2 mM external K⁺ from a holding voltage of −100 mV and step depolarizations from −100 mV to +80 mV (10 mV increments). **f)** Time constants of fast inactivation (τ) plotted against test voltage in 2 mM external K⁺ for Shaker N482L compared to WT-Shaker expressed by RNA injection. A single exponential fit to current traces like those in panel a was

used to obtain values of τ. Data for WT-Shaker are from Fig. 1f and for N482L n = 5 cells in 4 independent experiments. **g)** Family of current traces to measure recovery from inactivation for Shaker N482L expressed by RNA injection. Holding voltage was −100 mV and initial step depolarizations were to +30 mV for 60 ms followed by varying amounts of time for recovery at −100 mV (initially 2 ms and increased in 5 ms intervals) before eliciting a second step depolarization to +30 mV. External K⁺ was 100 mM. **h)** Fraction of current recovered at −100 mV ($f_R$) as a function of recovery time for Shaker N482L and WT-Shaker expressed by RNA injection. Data for WT-Shaker are from Fig. 1h and for N482L n = 4 cells in 4 independent experiments for 2 mM external K⁺ and n = 5 in 3 independent experiments for 100 mM external K⁺. **i)** Current traces for Shaker N480W. Same protocol and conditions as panel e. **j)** Time constants of fast inactivation (τ) plotted against test voltage in 2 mM external K⁺ for Shaker N480W compared to WT-Shaker expressed by RNA injection. Data for WT-Shaker are from Fig. 1f and for N480W n = 4 cells in 2 independent experiments. **k)** Family of current traces to measure recovery from inactivation for Shaker N480W expressed by RNA injection. Same protocol and conditions as panel g. **l)** Fraction of current recovered at −100 mV ($f_R$) as a function of recovery time for Shaker N480W and WT-Shaker expressed by RNA injection. Data for WT-Shaker are from Fig. 1h and for N480W n = 3 cells in 2 independent experiments for 2 mM external K⁺ and n = 3 in 2 independent experiments for 100 mM external K⁺. P/−4 subtraction was used for all traces shown and red dotted line denotes zero current. Error bars are S.E.M.

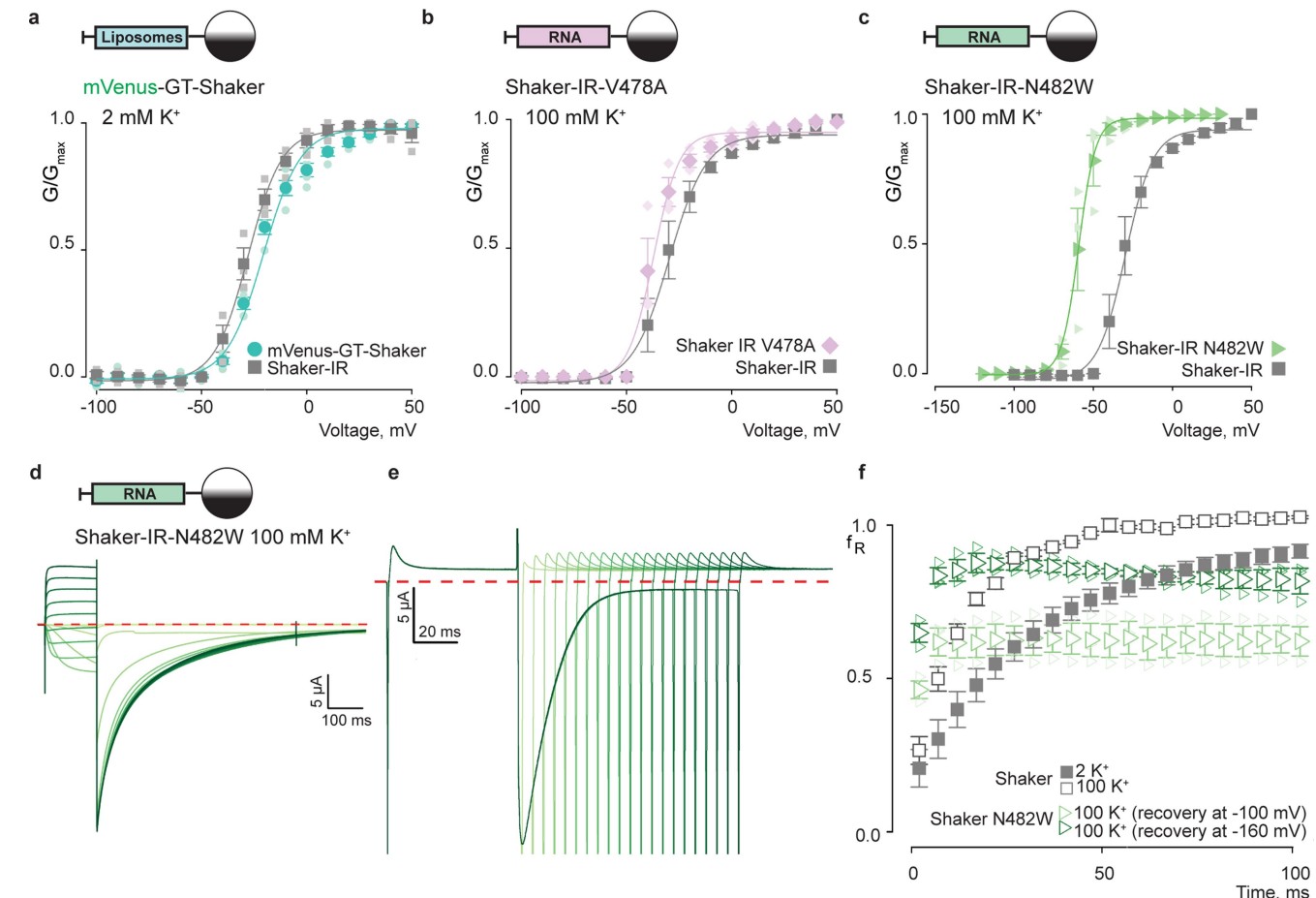

**Extended Data Fig. 9 | Functional properties of Shaker constructs.**
**a**) Normalized tail current voltage-activation (G-V) relations for mVenus-GT-Shaker expressed in oocytes by injecting liposomes containing reconstituted purified proteins expressed in HEK cells compared to Shaker-IR expressed by injection of RNA. In both cases, external $K^+$ was 2 mM, holding voltage was −100 mV and tail voltage was −50 mV. For Shaker-IR, n = 3 cells in 2 independent experiments and single Boltzmann fits yielded a $V_{1/2}$ of −29 ± 1.8 mV. For mVenus-GT-Shaker, n = 5 cells in 2 independent experiments and single Boltzmann fits yielded a $V_{1/2}$ of −25.6 ± 1.7 mV. **b**) Normalized tail current voltage-activation (G-V) relations for V478A Shaker-IR compared to Shaker-IR expressed by injection of RNA. In both cases, external $K^+$ was 100 mM, holding voltage was −100 mV and tail voltage was −50 mV. For Shaker-IR, n = 3 cells in 2 independent experiments and single Boltzmann fits yielded a $V_{1/2}$ of −30.1 ± 3.8 mV. For V478A Shaker-IR, n = 3 cells in 2 independent experiments and single Boltzmann fits yielded a $V_{1/2}$ of −37.0 ± 2.5 mV. **c**) Normalized tail current voltage-activation (G-V) relations for N482W Shaker-IR compared to Shaker-IR expressed by injection of RNA. For N482W, external $K^+$ was 100 mM, holding voltage was

−120 mV and tail voltage was −120 mV. Data for Shaker-IR are from panel b and for N482W Shaker-IR, n = 3 cells in 2 independent experiments and single Boltzmann fits yielded a $V_{1/2}$ of −59.0 ± 3.3 mV. **d**) Current traces for N482W Shaker-IR expressed in oocytes by injecting RNA recorded in 100 mM external $K^+$ using a holding voltage of −120 mV, step depolarizations from −100 mV to +30 mV (10 mV increments) and a tail voltage of −120 mV. **e**) Family of current traces to measure recovery from inactivation for Shaker N482W expressed by RNA injection. Holding voltage was −120 mV and initial step depolarizations were to +30 mV for 60 ms followed by varying amounts of time for recovery at −160 mV (initially 2 ms and increased in 5 ms intervals) before eliciting a second step depolarization to +30 mV. External $K^+$ was 100 mM. **f**) Fraction of current recovered at −100 or −160 mV ($f_R$) as a function of recovery time for Shaker N482W and WT-Shaker expressed by RNA injection. Data for WT-Shaker are from Fig. 1h and for N482W n = 3 cells in 2 independent experiments for recovery at −100 mV and n = 3 in 2 independent experiments for recovery at −160 mV, in both cases using 100 mM external $K^+$. P/−4 subtraction was used for all traces shown and red dotted line denotes zero current. Error bars are S.E.M.

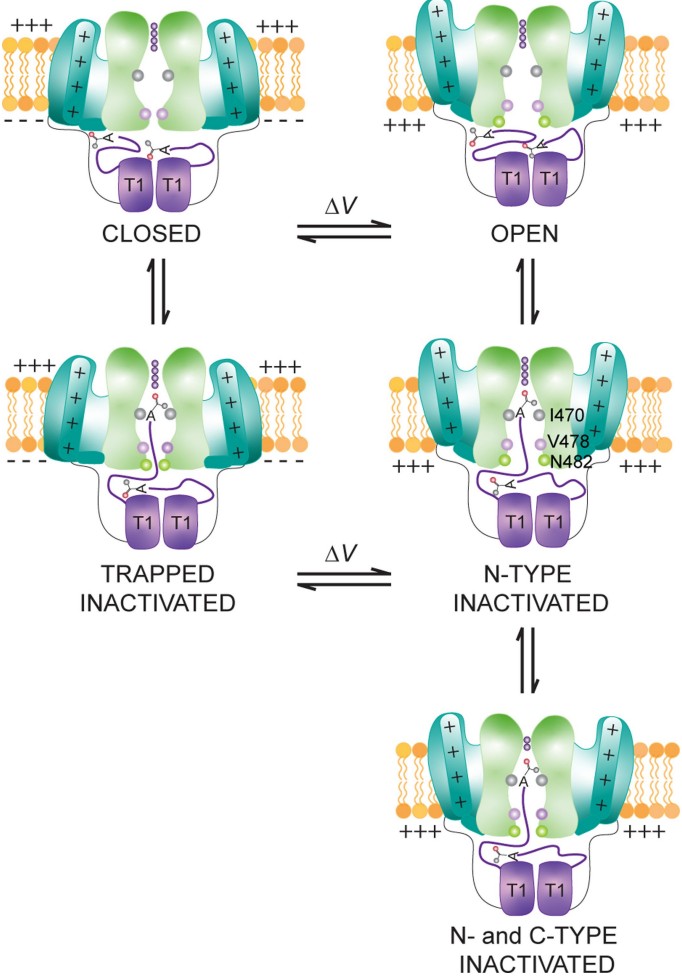

CLOSED

$\Delta V$

TRAPPED
INACTIVATED

N-TYPE
INACTIVATED

I470
V478
N482

$\Delta V$

N- and C-TYPE
INACTIVATED

**Extended Data Fig. 10 | Mechanism of fast inactivation.** Illustrations of the mechanism of fast N-type inactivation depicting key structural features and interactions with other states of the Shaker Kv channel. Under physiological conditions, the concentration of K[+] ions are low outside and high inside the cell. The internal pore of the channel is closed when the peripheral voltage-sensing domains are in resting conformations at negative membrane voltages and opens with membrane depolarization and activation of the voltage-sensing domains[16,39,40]. The uncharged N-terminus enters the open internal pore to diminish ion permeation[20–23], with an acetylated Ala on the immediate N-terminus interacting directly with I470, a key residue that is modified by RNA editing to regulate N-type inactivation[14]. V478 is a key residue forming the activation gate when the internal pore is closed[16,39,40], while in the N-type inactivated state it interacts intimately with L7, one of the most critical residues within the N-terminus that stabilizes the N-type inactivated state[20]. N-type inactivation promotes C-type inactivation of the ion selectivity filter by lowering the concentration of K[+] on the internal side of the selectivity filter and holding the activation gate open[46]. C-type inactivation also stabilizes the N-type inactivated state. Raising external K[+] concentrations speeds recovery from inactivation by shifting the selectivity filter from the C-type inactivated state into a conducting state. Strong hyperpolarization when channels are N-type inactivated can trap the inactivation particle inside the pore[51] as the internal pore closes off around an extended conformation of the N-terminus bound within the pore.

**Extended Data Table 1 | Cryo-EM data collection, refinement and validation statistics**

| | C-terminal mVenus tagged Shaker (EMD-49336) (PDB 9NEU) T1 domain / (EMD-49333) (PDB 9NES) TM domain | GT-Shaker class A (EMD-49311) (PDB 9NEI) | AcA-EI-Shaker class C (EMD-49308) (PDB 9NEG) | AcA-EI Shaker with free peptide (EMD-49305) (PDB 9NEC) (conformation A)/ (PDB 9NED) (conformation B) |
|---|---|---|---|---|
| **Data collection and processing** | | | | |
| Magnification | 105,000x | 105,000x | 105,000x | 105,000x |
| Voltage (kV) | 300 | 300 | 300 | 300 |
| Electron exposure (e–/Å²) | 52 | 52 | 52 | 52 |
| Defocus range (μm) | -0.5 ~ -1.5 | -0.5 ~ -1.5 | -0.5 ~ -1.5 | -0.5 ~ -1.5 |
| Pixel size (Å) | 0.415 (Sup res.) | 0.415 (Sup res.) | 0.415 (Sup res.) | 0.415 (Sup res.) |
| Symmetry imposed | C4 | C1 | C1 | C1 |
| Initial particle images (no.) | 11,631,163 | 9,150,671 | 6,622,267 | 12,854,356 |
| Final particle images (no.) | 325,375 | 141,882 | 9,0434 | 109,792 |
| Map resolution (Å) | 2.98 (T1)/2.67 (TM) | 2.94 | 3.17 | 3.05 |
| FSC threshold | 0.143 | 0.143 | 0.143 | 0.143 |
| **Refinement** | | | | |
| Initial model used (PDB code) | AlphaFold/8TEO | 8TEO | 8TEO | 8TEO |
| Model resolution (Å) | 2.9(9NEU)/2.6(9NES) | 3.0 | 3.0 | 3.1(9NEC)/3.1(9NED) |
| FSC threshold | 0.143 | 0.143 | 0.143 | 0.143 |
| Map sharpening $B$ factor (Å²) | -161.6/-126.4 | -30 | -20 | -40 |
| Model composition | | | | |
| Non-hydrogen atoms | 3648/6654 | 7056 | 10596 | 10239/10239 |
| Protein residues | 432/800 | 855 | 1276 | 1287/1288 |
| Ligands | 0/34 | 34 | 34 | 34/34 |
| $B$ factors (Å²) | | | | |
| Protein | 48.19/45.80 | 109.57 | 116.05 | 103.64/103.60 |
| Ligand | --/45.33 | 103.70 | 81.17 | 78.00/78.03 |
| R.m.s. deviations | | | | |
| Bond lengths (Å) | 0.003/0.007 | 0.004 | 0.003 | 0.003/0.003 |
| Bond angles (°) | 0.632/0.692 | 0.663 | 0.607 | 0.611/0.610 |
| Validation | | | | |
| MolProbity score | 1.84/1.34 | 1.43 | 1.83 | 1.69/1.70 |
| Clashscore | 7.01/6.15 | 7.56 | 9.12 | 8.08/8.38 |
| Poor rotamers (%) | 0/0 | 0 | 0 | 0/0 |
| Ramachandran plot | | | | |
| Favored (%) | 92.79/98.44 | 97.93 | 95.10 | 96.28/96.36 |
| Allowed (%) | 7.21/1.56 | 2.07 | 4.90 | 3.72/3.64 |
| Disallowed (%) | 0/0 | 0 | 0 | 0/0 |

**Extended Data Table 2 | Cryo-EM data collection, refinement and validation statistics**

| | GT-Shaker T1 domain (EMD-49312) | GT-Shaker TM domain (EMD-49331) | GT-Shaker Class B (EMD-49337) | GT-Shaker Class C (EMD-49338) | AcA-EI-shaker Class A (EMD-49356) | AcA-EI-shaker Class B (EMD-49365) |
|---|---|---|---|---|---|---|
| **Data collection and processing** | | | | | | |
| Magnification | | | 105,000x | | | |
| Voltage (kV) | | | 300 | | | |
| Electron exposure (e–/$\text{Å}^2$) | | | 52 | | | |
| Defocus range (μm) | | | -0.5 ~ -1.5 | | | |
| Pixel size (Å) | | | 0.415 (Sup res.) | | | |
| Symmetry imposed | C4 | C4 | C1 | C1 | C1 | C1 |
| Initial particle images (no.) | | 9,150,671 | | | 6,622,267 | |
| Final particle images (no.) | 1,417,462 | 1,417,462 | 282,712 | 142,153 | 100,375 | 102,018 |
| Map resolution (Å) | 2.31 | 2.35 | 2.79 | 3.01 | 2.92 | 3.01 |
| FSC threshold | 0.143 | 0.143 | 0.143 | 0.143 | 0.143 | 0.143 |

# Reporting Summary

## Statistics

For all statistical analyses, confirm that the following items are present in the figure legend, table legend, main text, or Methods section.

| n/a | Confirmed | |
|---|---|---|
| ☐ | ☒ | The exact sample size (*n*) for each experimental group/condition, given as a discrete number and unit of measurement |
| ☐ | ☒ | A statement on whether measurements were taken from distinct samples or whether the same sample was measured repeatedly |
| ☒ | ☐ | The statistical test(s) used AND whether they are one- or two-sided *Only common tests should be described solely by name; describe more complex techniques in the Methods section.* |
| ☒ | ☐ | A description of all covariates tested |
| ☐ | ☒ | A description of any assumptions or corrections, such as tests of normality and adjustment for multiple comparisons |
| ☐ | ☒ | A full description of the statistical parameters including central tendency (e.g. means) or other basic estimates (e.g. regression coefficient) AND variation (e.g. standard deviation) or associated estimates of uncertainty (e.g. confidence intervals) |
| ☒ | ☐ | For null hypothesis testing, the test statistic (e.g. *F*, *t*, *r*) with confidence intervals, effect sizes, degrees of freedom and *P* value noted *Give P values as exact values whenever suitable.* |
| ☒ | ☐ | For Bayesian analysis, information on the choice of priors and Markov chain Monte Carlo settings |
| ☒ | ☐ | For hierarchical and complex designs, identification of the appropriate level for tests and full reporting of outcomes |
| ☒ | ☐ | Estimates of effect sizes (e.g. Cohen's *d*, Pearson's *r*), indicating how they were calculated |

*Our web collection on statistics for biologists contains articles on many of the points above.*

## Software and code

Policy information about availability of computer code

| Data collection | SerialEM 3.8.1 for cryo-EM and pClamp 10.7 for electrophysiology |
|---|---|
| Data analysis | RELION 4.0, cryoSPARC v4.5.3, MotionCor2 1.3.1, CTFFIND4 and Gautomatch 0.56 for cryo-EM. Phenix v1.19.1 and WinCoot 0.9.8.95 for model building. PyMOL 2.4.1, UCSF Chimera 1.15rc and ChimeraX 1.8 for displaying maps and models in figures. pClamp 10.7 and Origin 2023b for electrophysiology. |

For manuscripts utilizing custom algorithms or software that are central to the research but not yet described in published literature, software must be made available to editors and reviewers. We strongly encourage code deposition in a community repository (e.g. GitHub). See the Nature Portfolio guidelines for submitting code & software for further information.

## Data

Policy information about availability of data

All manuscripts must include a data availability statement. This statement should provide the following information, where applicable:
- Accession codes, unique identifiers, or web links for publicly available datasets
- A description of any restrictions on data availability
- For clinical datasets or third party data, please ensure that the statement adheres to our policy

All data needed to evaluate the conclusions in the paper are present in the paper. Maps for Shaker Kv channel constructs have been deposited in the Electron Microscopy Data Bank (EMDB) under accession codes EMD-49333 for Shaker TM domain in C4, EMD-49336 for Shaker T1 domain in C4, EMD-49331 for GT-Shaker

# Research involving human participants, their data, or biological material

Policy information about studies with human participants or human data. See also policy information about sex, gender (identity/presentation), and sexual orientation and race, ethnicity and racism.

| | |
|---|---|
| Reporting on sex and gender | n/a |
| Reporting on race, ethnicity, or other socially relevant groupings | n/a |
| Population characteristics | n/a |
| Recruitment | n/a |
| Ethics oversight | n/a |

Note that full information on the approval of the study protocol must also be provided in the manuscript.

# Field-specific reporting

Please select the one below that is the best fit for your research. If you are not sure, read the appropriate sections before making your selection.

☒ Life sciences    ☐ Behavioural & social sciences    ☐ Ecological, evolutionary & environmental sciences

For a reference copy of the document with all sections, see nature.com/documents/nr-reporting-summary-flat.pdf

# Life sciences study design

All studies must disclose on these points even when the disclosure is negative.

| | |
|---|---|
| Sample size | Statistical methods were not used to determine sample size. Sample size for cryo-EM studies was determined by availability of microscope time and to ensure we obtain sufficient resolution for model building. Sample size for electrophysiological studies was determined empirically by comparing individual measurements with population data obtained under differing conditions until convincing differences or lack thereof were evident. |
| Data exclusions | For electrophysiological experiments, exploratory experiments were undertaken with varying ionic conditions and voltage-clamp protocols to define ideal conditions for measurements reported in this study. Although these preliminary experiments are consistent with the results we report, they were not included in our analysis due to varying experimental conditions. Once ideal conditions were identified, electrophysiological data were collected for control and mutant constructs until convincing trends in population datasets were obtained. Individual cells were also excluded if cells exhibited excessive initial leak currents at the holding voltage (>0.5 μA), if currents arising from expressed channels were too small (<0.5 μA), making it difficult to distinguish the activity of expressed channels from endogenous channels, or if currents arising from expressed channels were too large, resulting in substantial voltage errors or changes in the concentration of ions in either intracellular or extracellular solutions. |
| Replication | Information on sample size is provided in figure legends throughout the manuscript. |
| Randomization | Randomization was not used in this study, because these experiments did not involve allocating discrete samples to experimental groups. |
| Blinding | Blinding was not used in this study, because the knowledge of the sample doesn't affect the measurement of the datasets. |

# Reporting for specific materials, systems and methods

We require information from authors about some types of materials, experimental systems and methods used in many studies. Here, indicate whether each material, system or method listed is relevant to your study. If you are not sure if a list item applies to your research, read the appropriate section before selecting a response.

## Materials & experimental systems

| n/a | Involved in the study |
|-----|----------------------|
| ☒ | ☐ Antibodies |
| ☐ | ☒ Eukaryotic cell lines |
| ☒ | ☐ Palaeontology and archaeology |
| ☐ | ☒ Animals and other organisms |
| ☒ | ☐ Clinical data |
| ☒ | ☐ Dual use research of concern |
| ☒ | ☐ Plants |

## Methods

| n/a | Involved in the study |
|-----|----------------------|
| ☒ | ☐ ChIP-seq |
| ☒ | ☐ Flow cytometry |
| ☒ | ☐ MRI-based neuroimaging |

## Eukaryotic cell lines

Policy information about cell lines and Sex and Gender in Research

| | |
|---|---|
| Cell line source(s) | Sf9 and tsA201 cells were originally obtained from Thermo Fischer and Sigma-Aldrich, respectively. |
| Authentication | Cell lines used were not authenticated. |
| Mycoplasma contamination | Mycoplasma contamination was tested and found to be negative |
| Commonly misidentified lines (See ICLAC register) | commonly misidentified cell lines were not used in this study |

## Animals and other research organisms

Policy information about studies involving animals; ARRIVE guidelines recommended for reporting animal research, and Sex and Gender in Research

| | |
|---|---|
| Laboratory animals | female Xenopus laevis frogs, 1-2 years of age, obtained from Xenopus I |
| Wild animals | no wild animals were used in the study |
| Reporting on sex | Female |
| Field-collected samples | no field collected samples were used in the study |
| Ethics oversight | The animal care and experimental procedures were performed in accordance with the Guide for the Care and Use of Laboratory Animals and were approved by the Animal Care and Use Committee of the National Institute of Neurological Disorders and Stroke (animal protocol number 1253). |

Note that full information on the approval of the study protocol must also be provided in the manuscript.

