## [Peer Review file · Nature]

Structural basis of fast N-type inactivation in Kv channels

Corresponding Author: Dr Kenton Swartz

Version 0:

Reviewer comments:

Referee #1

(Remarks to the Author)

The manuscript by Tan, Fernández-Mariño, and colleagues advances the mechanistic understanding of fast-inactivation of Shaker potassium channels. The authors use cryo-EM, mass spectrometry, and functional measurements from heterologously expressed channels to make a number of surprising insights into a channel that has been a workhorse in biophysical studies for decades. The significance of this work is further compounded by the recent observation that Shaker family K⁺ channels, evolutionarily speaking, have outlived the modern neurological system. That is, this family of potassium channels has been found in choanoflagellates (PMID: 39018191) and recently in ctenophores (PMID: 4010064), which represent one of the oldest living branches of the animal kingdom. In providing the structure of a prototypical Shaker channel, the authors have on one level, identified another detail in channel function (fast inactivation), but on a much broader level, these data provide snapshots into the pre-nervous system, evolutionary past.

The authors first demonstrate that TEV cleavage produces a vestigial N-terminal dipeptide "GT" which slows entry to, and speeds exit from, the Shaker fast-inactivated state. Similarly, an N-terminal Venus peptide completely abolishes fast-inactivation. High-resolution structures of these GT channels have a poor density within the pore of the putative N-terminal peptide region. The authors use a clever masking approach to identify three subclasses of structural poses, with the A class being assigned as the likely N-terminally occluded pore i.e. fast-inactivated state. The authors then use mass spec to show that the N-terminus of Shaker is naturally Met-cleaved and acetylated, and this feature is termed the AcA Shaker channel. This is a nice finding that will resonate with many scientists in the ion channel community. Next the authors solve the structures of the AcA channel with two mutations which enhance fast-inactivation (AcA-EI-Shaker) as well as a condition that contains free N-terminal EI peptide. These structures reveal a surprising discovery that the N-terminal charged residues E12K and D13K sit outside of the pore, just below the inner bundle crossing closure. This is unexpected because previous studies using increased external potassium to manipulate fast-inactivation rates have evoked an electrostatic repulsion between these charges and permeant ions. These structures suggest that this mechanism would be unrealistic. For me, the excitement of the study weakened significantly with the last two figures. Here the authors use site-directed mutagenesis on Shaker channels expressed in oocytes to advance the understanding of fast inactivation and how it interacts with the selectivity filter. The experiments are clever and are of high quality but ultimately fall short of "nailing" the question of how external K⁺ ions regulate this form of inactivation.

The paper is clearly written for a general audience and accurately referenced. Overall, this is extremely well done science on a topic that will broadly resonate with the ion channel community.

Additional comments in no particular order of significance:

As mentioned by the authors, N-terminal acetylation is a known modification thus its discovery on Shaker will be of interest to the many labs that have used this channel to study channel mechanisms. That being said, it is not biologically novel and that this one insight takes up an entire figure seems unnecessary.

Many Kv and Nav structures appear to have a detergent or similar molecule in their pore. Can the authors be entirely sure that the class A particles are 100% peptide occluded and not detergent bound within the pore?

I take minor issue with nonchuku naming. Is this really that much different that ball-and-chain fast inactivation? What sorts of structural and mechanistic differences exist between the "ball" peptide and nonchuku? And a minor detail is that the pore-bound N-terminal peptide seems to be mostly unstructured, opposed to a rigid α -helix. The latter, helical structure would be more in line with the nonchuku reference.

Referee #2

(Remarks to the Author)

This is a very nice manuscript describing a molecular and structural basis for fast N-type inactivation in Shaker-like Kv channels. It has long been thought that Shaker and certain other Kv channels undergo fast inactivation due to their N-terminal region physically occluding the open pore. Work from others spanning decades have indicated that N-type inactivation can be likened to a ball-and-chain type of mechanism wherein the ball plugs the pore shortly after it opens. In this work, the authors determine cryo-EM structures of Shaker that indicate that the inactivation domain binds in the open pore to occlude ion permeation. In addition, the authors make the surprising discoveries that the N-terminal methionine residue of inactivation domain is removed and that the second amino acid undergoes acetylation. Their structural and functional studies suggest that the acetylated N-terminus binds deep within the pore, below the selectivity filter, where it would block ion conduction. A structural basis for N-type inactivation in Kv channels has long been sought, which makes the structural studies performed here satisfying. As the authors indicate, there is strong evidence that the inactivation domain in the Shaker channel adopts an extended conformation when it binds in the pore to block ion conduction. Observing density for such a peptide presents a challenge for two reasons: 1) The channel has four-fold symmetry but only a single peptide is required to plug the channel – therefore, it is difficult to isolate a discrete conformation of the peptide relative to the symmetric arrangement of the pore. 2) The inactivation domain is comprised of amino acids that are relatively featureless (the amino acid sequence is AAVAGLY...), which could make it difficult to assign amino acids to observed density. The authors determine structures under three different conditions with density that can be ascribed to an inactivation domain present within the pore. In comparison to the surrounding helices of the channel, the density is weak and has less identifiable features. It is possible that the density represents an average of multiple conformations. The authors have modeled plausible conformations of the peptide that are consistent with the density, the pre-existing work of others, and their own mutagenesis based upon the structures. The authors perform beautiful electrophysiological studies that support the structural work. The main weakness of the study is that the density ascribed to the peptide is not very-well defined and is weak in comparison to the surrounding protein of the channel. As indicated above, this is to be expected given the limitations of observing an extended peptide within a symmetric pore. However, some of the conclusions such as which amino acids on the inactivation domain interact with particular amino acids on the sides of the channel cannot be supported by the density alone. To their credit, the authors make excellent use of electrophysiological studies to evaluate some potential interactions. In aggregate, the conclusions are sound and structurally satisfying: the inactivation domain in Shaker binds within the pore in an extended conformation to block ion permeation.

Specific points

The authors put their work in the context of other studies on voltage gated cation channels and inactivation processes. The beginning part of the manuscript reads like there is a controversy regarding the inactivation mechanisms in cation channels, while the emerging theme is that different channels inactivate through different processes and that the ball-and-chain (or nunchaku) mechanism does not apply to all. This could be made clearer for the general reader.

The three structures with the inactivation domain are modeled to have slightly different peptide conformations (EI-shaker without peptide class C, EI-shaker with peptide, and GT-shaker class A). This could be made clearer, and it highlights the potential that the domain does not occupy a discrete conformation.

An example of the density for the inactivation domain is included on the attached figure. From this, one can appreciate that there is a degree of uncertainty with regard to amino acid register and position. This uncertainty should be presented more clearly. Of course, one would always like more well-defined density, but this may be difficult to achieve. In the models for the inactivation peptide, a couple residues are borderline-outliers on a Ramachandran plot, but this should be easy to fix.

Pg. 6. The authors discuss a potential interaction between Y8 and N482. From inspection of the models, this interaction appears to only be modeled in one of these structures (GT-shaker class A). In another structure (EI-shaker class C), these amino acids are approximately 8 Å apart.

Pg. 7. (A speculation) Regarding the coupling between C-type inactivation and N-type inactivation, it seems possible to this reviewer that potassium ions emerging from the bottom of the filter could destabilize the binding of the inactivation domain by van der Waals and molecular crowding due to the size of hydrated potassium ions.

The structures suggest a possible interaction between V4 and V474. If this reviewer had to guess, this would be one of the most energetically significant interactions. It could be interesting to look at mutations of V474 together with mutations of V4 in a double mutant-cycle type of approach (e.g. PMID 9080186). Double mutant cycle approaches could also be useful for supporting interactions such as that proposed between Y8 and N482. In the current work, the authors mutate the channel residues (e.g. N482) but not the residues on the inactivation domain.

Minor point. In the cryo-EM maps, there is strong density for the connection between S2 and S3, but this region is not included in the atomic model. Likewise, there is strong density preceding S1 that could be modeled. This does not detract from the main point of the manuscript because these regions are part of the voltage sensor, which is not the focus of the study.

Pg. 6. Minor point. In relation to positions 12 and 13, the authors state: "These two residues are clearly positioned outside

the pore". The "clearly" wording could be improved: the density for these amino acids is weak in the cryo-EM maps. I do agree with the authors assessment that these residues are located outside of the pore.

Of course, it would be nice to see a structure of the wild-type Shaker channel with an intact N-terminal inactivation domain bound within the pore. However, the properties of the EI-Shaker mutant are well-documented and are expected to reveal the same principles for inactivation as the wild-type inactivation domain.

Version 1:

Reviewer comments:

Referee #2

(Remarks to the Author)

The authors have sufficiently addressed my comments. I have only a few remaining comments that might be worth considering.

The current title: "Mechanism of fast N-type inactivation in Kv channels" could be a bit misleading. The mechanism of fast N-type inactivation in Kv channels has long been established to involve the ball-and-chain mechanism that the authors visualize. The authors determine a structural basis for this mechanism. Therefore, a possible suggestion for a revised title is "Structural basis of fast N-type inactivation in Kv channels"

The authors mention a few times that proteolysis occurred for the N-terminal peptide in some cases. It would be nice to include data showing this to be the case. (Perhaps this data was included in the MS data, but it was difficult for me to discern.) Would an SDS-PAGE gel also be useful?

Pg. 8. "stabilizes the inactivated state by interacting directly with the N-terminus at the mouth of the inner pore." I was a bit confused by the phrasing "N-terminus" – do the authors wish to refer to a portion of the N-terminal peptide, or the N-terminal (acylated) residue?

Manuscript 2025-02-05107

We greatly appreciate the thoughtful guidance of the editor and reviewers. We have now prepared a revised version for further consideration. The comments of the reviewers are shown below in black text and our responses and descriptions of the changes made are shown in blue text. We have also included a word document with "track changes" turned on so everyone can see where the text has changed.

Referee #1 (Remarks to the Author):

The manuscript by Tan, Fernández-Mariño, and colleagues advances the mechanistic understanding of fast-inactivation of Shaker potassium channels. The authors use cryo-EM, mass spectrometry, and functional measurements from heterologously expressed channels to make a number of surprising insights into a channel that has been a workhorse in biophysical studies for decades. The significance of this work is further compounded by the recent observation that Shaker family K⁺ channels, evolutionarily speaking, have outlived the modern neurological system. That is, this family of potassium channels has been found in choanoflagellates (PMID: 39018191) and recently in ctenophores (PMID: 40100064), which represent one of the oldest living branches of the animal kingdom. In providing the structure of a prototypical Shaker channel, the authors have on one level, identified another detail in channel function (fast inactivation), but on a much broader level, these data provide snapshots into the pre-nervous system, evolutionary past.

This is a really thoughtful point, and we now mention this at the end of the discussion where those papers are cited.

The authors first demonstrate that TEV cleavage produces a vestigial N-terminal dipeptide "GT" which slows entry to, and speeds exit from, the Shaker fast-inactivated state. Similarly, an N-terminal Venus peptide completely abolishes fast-inactivation. High-resolution structures of these GT channels have a poor density within the pore of the putative N-terminal peptide region. The authors use a clever masking approach to identify three subclasses of structural poses, with the A class being assigned as the likely N-terminally occluded pore i.e. fast-inactivated state. The then authors use mass spec to show that the N-terminus of Shaker is naturally Met-cleaved and acetylated, and this feature is termed the AcA Shaker channel. This is a nice finding that will resonate with many scientists in the ion channel community. Next the authors solve the structures of the AcA channel with two mutations which enhance fast-inactivation (AcA-EI-Shaker) as well as a condition that contains free N-terminal EI peptide. These structures reveal a surprising discovery that the N-terminal charged residues E12K and D13K sit outside of the pore, just below the inner bundle crossing closure. This is unexpected because

previous studies using increased external potassium to manipulate fast-inactivation rates have evoked an electrostatic repulsion between these charges and permeant ions. These structures suggest that this mechanism would be unrealistic.

For me, the excitement of the study weakened significantly with the last two figures. Here the authors use site-directed mutagenesis on Shaker channels expressed in oocytes to advance the understanding of fast inactivation and how it interacts with the selectivity filter. The experiments are clever and are of high quality but ultimately fall short of “nailing” the question of how external K⁺ ions regulate this form of inactivation.

The paper is clearly written for a general audience and accurately referenced. Overall, this is extremely well done science on a topic that will broadly resonate with the ion channel community.

We really appreciate these supportive comments and have reworked the last section of the results section to address this reviewer's diminished enthusiasm for that part of the study. We more openly discuss nuances about the influence of external K⁺ on recovery from inactivation and have rewritten the results section concerning the interaction between N482 and Y8.

Additional comments in no particular order of significance:

As mentioned by the authors, N-terminal acetylation is a known modification thus its discovery on Shaker will be of interest to the many labs that have used this channel to study channel mechanisms. That being said, it is not biologically novel and that this one insight takes up an entire figure seems unnecessary.

We appreciate this perspective. As the reviewer states, this is an important result that will be of interest to those studying related mechanisms of inactivation in many different channels because knowing the identity of the blocking particle is quite fundamental. We also really like this experiment because injection of oocytes with the purified channel enabled us to functionally characterize the same protein samples used for structure determination, something that is rarely achievable in this field. While N-terminal modifications of proteins is of course very common, it had never been studied for a protein where the N-terminus serves such a critical role as it does in N-type inactivation in Shaker.

Many Kv and Nav structures appear to have a detergent or similar molecule in their pore. Can the authors be entirely sure that the class A particles are 100% peptide occluded and not detergent bound within the pore?

In many instances where elongated densities have been seen in the internal pore of Kv channels, the structures have been solved in detergent solution, so it seems possible

they are in fact detergent molecules. We have also seen this in Kv channel structures we solved in detergents, however, in the current study all samples were reconstituted into nanodiscs and purified using detergent-free buffer, with most detergent removed during size-exclusion chromatography. In nanodiscs, we have also observed elongated densities within the internal pore under specific conditions, but we think it is unlikely that detergents could have remained bound within the pore during purification of the nanodiscs. In our previous published structures of Shaker-IR in a conducting state and Shaker-W434F in a C-type inactivated state, we only observed four elongated densities occupying the internal pore in the conducting state. Our experience is that these four elongated densities can be resolved into shapes similar to water-cages when we were able to improve the resolution of around 2.7 Å or better. While additional efforts would be needed to fully understand these densities, we think it is unlikely they are detergents under the conditions we have been working. In addition, for both AcA-Shaker and AcA-EI-Shaker datasets, we consistently observed one major class (~50% of particles) that showed no density within the internal pore and a dilated selectivity filter. The absence of any density in these datasets and the fact that they have C-type inactivated filters, make us confident that the densities we do see within the internal pore in the two relevant classes, correspond to channels with the N-terminal inactivation peptides inserted into the internal pore.

I take minor issue with nonchuku naming. Is this really that much different that ball-and-chain fast inactivation? What sorts of structural and mechanistic differences exist between the “ball” peptide and nonchuku? And a minor detail is that the pore-bound N-terminal peptide seems to be mostly unstructured, opposed to a rigid α -helix. The latter, helical structure would be more line with the nonchuku reference.

This is a very reasonable point that was also echoed by the editor (and several colleagues). We have removed the nunchaku reference from the revised version of the manuscript.

Referee #2 (Remarks to the Author):

This is a very nice manuscript describing a molecular and structural basis for fast N-type inactivation in Shaker-like Kv channels. It has long been long thought that Shaker and certain other Kv channels undergo fast inactivation due to their N-terminal region physically occluding the open pore. Work from others spanning decades have indicated that N-type inactivation can be likened to a ball-and-chain type of mechanism wherein the ball plugs the pore shortly after it opens. In this work, the authors determine cryo-EM structures of Shaker that indicate that the inactivation domain binds in the open pore to occlude ion permeation. In addition, the authors make the surprising discoveries that the N-terminal methionine residue of inactivation domain is removed and that the

second amino acid undergoes acetylation. Their structural and functional studies suggest that the acetylated N-terminus binds deep within the pore, below the selectivity filter, where it would block ion conduction. A structural basis for N-type inactivation in Kv channels has long been sought, which makes the structural studies performed here satisfying. As the authors indicate, there is strong evidence that the inactivation domain in the Shaker channel adopts an extended conformation when it binds in the pore to block ion conduction. Observing density for such a peptide presents a challenge for two reasons: 1) The channel has four-fold symmetry but only a single peptide is required to plug the channel – therefore, it is difficult to isolate a discrete conformation of the peptide relative to the symmetric arrangement of the pore. 2) The inactivation domain is comprised of amino acids that are relatively featureless (the amino acid sequence is AAVAGLY...), which could make it difficult to assign amino acids to observed density. The authors determine structures under three different conditions with density that can be ascribed to an inactivation domain present within the pore. In comparison to the surrounding helices of the channel, the density is weak and has less identifiable features. It is possible that the density represents an average of multiple conformations. The authors have modeled plausible conformations of the peptide that are consistent with the density, the pre-existing work of others, and their own mutagenesis based upon the structures. The authors perform beautiful electrophysiological studies that support the structural work. The main weakness of the study is that the density ascribed to the peptide is not very-well defined and is weak in comparison to the surrounding protein of the channel. As indicated above, this is to be expected given the limitations of observing an extended peptide within a symmetric pore. However, some of the conclusions such as which amino acids on the inactivation domain interact with particular amino acids on the sides of the channel cannot be supported by the density alone. To their credit, the authors make excellent use of electrophysiological studies to evaluate some potential interactions. In aggregate, the conclusions are sound and structurally satisfying: the inactivation domain in Shaker binds within the pore in an extended conformation to block ion permeation.

We appreciate the reviewer's supportive comments and below we address limitations arising from the scenario the reviewer describes. We completely agree that there are likely to be many conformations of the N-terminus engaged with the internal pore, but we also think many features of the density clearly support our assignments as corresponding to populated conformations. We are confident that careful inspection of our collective maps at different contours would support our assignments.

Specific points

The authors put their work in the context of other studies on voltage gated cation channels and inactivation processes. The beginning part of the manuscript reads like there is a controversy regarding the inactivation mechanisms in cation channels, while

the emerging theme is that different channels inactivate through different processes and that the ball-and-chain (or nunchaku) mechanism does not apply to all. This could be made clearer for the general reader.

We appreciate this comment and have edited the abstract and introduction to make this clearer.

The three structures with the inactivation domain are modeled to have slightly different peptide conformations (EI-shaker without peptide class C, EI-shaker with peptide, and GT-shaker class A). This could be made clearer, and it highlights the potential that the domain does not occupy a discrete conformation.

We agree and now explicitly make this point in the results section when presenting the structures of AcA-EI Shaker alone and with additional free peptide.

An example of the density for the inactivation domain is included on the attached figure. From this, one can appreciate that there is a degree of uncertainty with regard to amino acid register and position. This uncertainty should be presented more clearly. Of course, one would always like more well-defined density, but this may be difficult to achieve. In the models for the inactivation peptide, a couple residues are borderline-outliers on a Ramachandran plot, but this should be easy to fix.

We appreciate this point and have revised our model of AcA-EI-Shaker to move L7 more in line with the map as suggested by the reviewer in the figure they provided. We have also addressed the outliers in the Ramachandran plot and overall have tried to be more nuanced in our presentation in the revised manuscript.

During data processing, we observed that a single round of 3D classification using the L-shaped mask couldn't fully separate distinct conformations. Further subclassification of selected particles could improve the separation, but never fully, with the cost of reduced resolution due to the smaller number of particles in each class. This necessitated a balance between classification accuracy and resolution.

Because the transmembrane region behaves as a relatively rigid body, while the N-terminal region includes a dominant conformation mixed with a small number of particles from alternative conformations, the density for the N-terminal region appeared weaker than that of the transmembrane region. As a result, we used a lower threshold to build the N-terminal model than was used for the transmembrane domain.

The example shown in the attachment from this reviewer was from the EI-Shaker dataset, which was the most challenging of the three datasets during data processing. Due to proteolysis, approximately 50% of particles lacked any N-terminal density, and about 10% showed a cleaved peptide fragment occupying the pore entrance, so the

number of particles remaining with intact N-termini was insufficient to fully separate distinct N-type inactivation conformations with high resolution.

When we lowered the density threshold in maps for AcA-EI-Shaker, we could see that the observed N-terminal density was an average of two conformations—one dominant and one weaker (especially the beginning region AcA-A2A3V4A5G6), making residue assignment challenging.

To make sure the assignment was correct in EI-Shaker, we also collected an EI-Shaker dataset in the presence of free N-terminal peptide to drive all the particles into a fully inactivated conformation, which gave us better separation in the region AcA-A2A3V4A5G6. Our modeling strategy for N-terminal domain began by fitting the first residue and those with large side chains, such as L7 and Y8, into the density, followed by fitting residues with shorter sidechains. We then validated the models using shared features across the three fully inactivated structures (GT-Shaker, EI-Shaker, and EI-Shaker with free peptide).

We really appreciate the reviewers prompt to discuss the uncertainty of residue assignments and we have carefully edited the results section to provide a clearer description of the challenges and how we dealt with them.

Pg. 6. The authors discuss a potential interaction between Y8 and N482. from inspection of the models, this interaction appears to only be modeled in one of these structures (GT-shaker class A). In another structure (EI-shaker class C), these amino acids are approximately 8 Å apart.

It is true that Y8 and N482 are positioned nearby in GT-Shaker and AcA-EI-Shaker with free peptide, but far apart in the structure of AcA-EI Shaker because Y8 is displaced by the second incoming N-terminal peptide. That Y8 can be readily displaced is consistent with our functional data on truncating N482 to Ala, which suggests that N482 and Y8 are unlikely to interact in an energetically meaningful way in the WT channel. However, the interesting point is that mutating N482 to Ala, Leu and Trp progressively stabilized the N-type inactivated state, consistent with hydrophobic substitutions at this position interacting with Y8 (or L7) positioned at the inner mouth of the pore nearby to N482. We have extensively rewritten this section to hopefully make these points much more clearly.

Pg. 7. (A speculation) Regarding the coupling between C-type inactivation and N-type inactivation, it seems possible to this reviewer that potassium ions emerging from the bottom of the filter could destabilize the binding of the inactivation domain by van der Waals and molecular crowding due to the size of hydrated potassium ions.

We agree with this speculation. While we think we provide clear evidence that C-type inactivation stabilizes the N-type inactivated state and that the former is disfavored as

external K⁺ increases, it is clear that external K⁺ and negative voltage speed recovery from N-type inactivation to some extent when C-type inactivation is disfavored with the T449V mutation. We have revised the relevant sections of the manuscript to make this clear and to speculate on the mechanisms raised by this reviewer.

The structures suggest a possible interaction between V4 and V474. If this reviewer had to guess, this would be one of the most energetically significant interactions. It could be interesting to look at mutations of V474 together with mutations of V4 in a double mutant-cycle type of approach (e.g. PMID 9080186). Double mutant cycle approaches could also be useful for supporting interactions such as that proposed between Y8 and N482. In the current work, the authors mutate the channel residues (e.g. N482) but not the residues on the inactivation domain.

This is an excellent point. In the inactivation domain had been previously studied extensively by the laboratories of Rick Aldrich and Rod MacKinnon and we have emphasized those contributions in the revised manuscript. MacKinnon used mutant cycle analysis on N-type inactivation in Kv1.4 produced by the β 12 subunit and showed strong coupling between a Val at position 3 in their N-terminal peptide and the equivalent of V474 (V562), which we now mention in the revised manuscript. We also rewrote the section of the results concerning N482 and Y8 to be clearer what we were trying to accomplish with mutations at N482.

Minor point. In the cryo-EM maps, there is strong density for the connection between S2 and S3, but this region is not included in the atomic model. Likewise, there is strong density preceding S1 that could be modeled. This does not detract from the main point of the manuscript because these regions are part of the voltage sensor, which is not the focus of the study.

For the density preceding S1, we modeled 10 more residues into the density as suggested. In our previous structures where we imposed C4 symmetry, we could not see density for the S2-S3 linker region and therefore did not model that region. We originally attempted to model the density connecting S2 and S3 in our current C1 maps, but we found that the density did not provide sufficient space to fit all the missing residues, nor did it exhibit large sidechain features to indicate the direction of the main chain. We therefore decided to leave the density between S2 and S3 unmodeled.

Pg. 6. Minor point. In relation to positions 12 and 13, the authors state: "These two residues are clearly positioned outside the pore". The "clearly" wording could be improved: the density for these amino acids is weak in the cryo-EM maps. I do agree with the authors assessment that these residues are located outside of the pore.

We have reworded these statements. Thanks,

Of course, it would be nice to see a structure of the wild-type Shaker channel with an intact N-terminal inactivation domain bound within the pore. However, the properties of the EI-Shaker mutant are well-documented and are expected to reveal the same principles for inactivation as the wild-type inactivation domain.

We agree and did expend a great deal of energy trying to capture the N-type inactivated state in the WT channel.

Manuscript 2025-02-05107A

We greatly appreciate the thoughtful guidance of the editor and reviewers. We have now prepared a revised version for further consideration. The comments of the reviewers are shown below in black text and our responses and descriptions of the changes made are shown in blue text. The changes made to address the requests of the editor are as follows:

- 1) We have reduced the number of words in the main text from about 5100 to 4762, somewhat more than the 300-word reduction requested.
- 2) We have moved the original Fig. 6 into the Extended Data as Extended Data Fig. 10.
- 3) The title has been changed as requested. (59 characters with spaces)
- 4) The abstract has been reduced to 232 words, close to the request of 230 words.
- 5) We have reduced the references to 55 in the main text.
- 6) We shortened several of the figure legends in the main figures.
- 7) We have moved 3 Extended Figures into the Supplementary Information and we now have a total of 12 Extended Data items, including 10 Extended Figures and 2 Extended Data Tables.
- 8) We have updated or completed all of the requested forms (checklists and reporting summaries).

Referee #2 (Remarks to the Author):

The authors have sufficiently addressed my comments. I have only a few remaining comments that might be worth considering.

The current title: “Mechanism of fast N-type inactivation in Kv channels” could be a bit misleading. The mechanism of fast N-type inactivation in Kv channels has long been established to involve the ball-and-chain mechanism that the authors visualize. The authors determine a structural basis for this mechanism. Therefore, a possible suggestion for a revised title is “Structural basis of fast N-type inactivation in Kv channels”

We have changed the title as suggested.

The authors mention a few times that proteolysis occurred for the N-terminal peptide in some cases. It would be nice to include data showing this to be the case. (Perhaps this

data was included in the MS data, but it was difficult for me to discern.) Would an SDS-PAGE gel also be useful?

We did not collect MS data until after we suspected there was proteolysis due to challenges seeing density within the pore. Our SDS-PAGE indeed reveal the presence of proteolysis, but do not convincingly distinguish between proteolysis of the N- or other flexible region of the protein. In hindsight it would be nice to have that data as a counter example for when the protein was not proteolyzed, but we don't have any useful data to include.

Pg. 8. "stabilizes the inactivated state by interacting directly with the N-terminus at the mouth of the inner pore." I was a bit confused by the phrasing "N-terminus" – do the authors wish to refer to a portion of the N-terminal peptide, or the N-terminal (acylated) residue?

We have rephrased this sentence to be clear we are talking about interactions at the inner end of the pore and several hydrophobic residues (L7 and/or Y8) within the N-terminus.

That sentence now reads:

"That the N482W mutation so dramatically slows recovery from inactivation (**Fig. 5e-h**), therefore suggests that it likely stabilizes the inactivated state by interacting directly with hydrophobic residues (L7 and/or Y8) of the N-terminus."

El-shaker class C

K 7792 KB

Ala2

CA 72 ALA/I

Ala3

Val4

CA 74 VAL/I

CA 77 LEU/I
Leu7

Tyr8